# Sparser Block-Sparse Attention via Token Permutation

**Xinghao Wang** [1 2] **Pengyu Wang** [1 2] **Dong Zhang** [1] **Chenkun Tan** [1 2] **Shaojun Zhou** [1 2] **Zhaoxiang Liu** [3]
**Shiguo Lian** [3] **Fangxu Liu** [4] **Kai Song** [4] **Xipeng Qiu** [1 2 5]

## Abstract

Scaling the context length of large language models (LLMs) offers significant benefits but is computationally expensive. This expense stems primarily from the self-attention mechanism, whose $O(N^2)$ complexity with respect to sequence length presents a major bottleneck for both memory and latency. Fortunately, the attention matrix is often sparse, particularly for long sequences, suggesting an opportunity for optimization. Block-sparse attention has emerged as a promising solution that partitions sequences into blocks and skips computation for a subset of these blocks. However, the effectiveness of this method is highly dependent on the underlying attention patterns, which can lead to sub-optimal block-level sparsity. For instance, important key tokens for queries within a single block may be scattered across numerous other blocks, leading to computational redundancy. In this work, we propose Permuted Block-Sparse Attention (**PBS-Attn**), a plug-and-play method that leverages the permutation properties of attention to increase block-level sparsity and enhance the computational efficiency of LLM prefilling. We conduct comprehensive experiments on challenging long-context datasets, demonstrating that PBS-Attn consistently outperforms existing block-sparse attention methods in model accuracy and closely matches the full attention baseline. Powered by our custom permuted-FlashAttention kernels, PBS-Attn achieves an end-to-end speedup of up to $2.75\times$ in long-context prefilling, confirming its practical viability. Code available at https://github.com/xinghaow99/pbs-attn.

---

[1]Fudan University [2]OpenMOSS Team [3]China Unicom [4]ByteDance [5]Shanghai Innovation Institute. Correspondence to: Xipeng Qiu <xpqiu@fudan.edu.cn>.

*Proceedings of the 43rd International Conference on Machine Learning*, Seoul, South Korea. PMLR 306, 2026. Copyright 2026 by the author(s).

## 1. Introduction

Modern Large Language Models (LLMs) have demonstrated remarkable proficiency in handling long-context tasks (OpenAI, 2025; Gemini Team, Google, 2025; Anthropic, 2025), a capability fueled by advancements in infrastructure (Liu et al., 2023; Jin et al., 2024), training methodologies (Yang et al., 2025a), and novel positional embedding schemes (Su et al., 2024; Press et al., 2022; Peng et al., 2024). This progress enables models to process context windows spanning thousands or even millions of tokens, unlocking novel applications such as analyzing entire codebases, summarizing lengthy legal documents, and interpreting long-form video content.

However, this extended capability is constrained by prohibitive memory and computational overheads. This bottleneck primarily stems from the self-attention mechanism within the Transformer architecture (Vaswani et al., 2017). The necessity for each token to attend to all other tokens results in a computational complexity that scales quadratically with the input sequence length, posing a fundamental challenge to scalable and accessible long-context processing.

Hardware-aware optimizations, exemplified by FlashAttention (Dao et al., 2022), reduce memory overhead by tiling the sequence into blocks and performing an online softmax computation. This method avoids the materialization of the full attention matrix, thereby alleviating the memory overhead and efficiency constraints imposed by I/O limitations. Building directly upon this tiled approach, block-sparse attention further reduces computation by skipping the computation for certain blocks using a pre-computed sparse block mask (Dao et al., 2022; Jiang et al., 2024; Lai et al., 2025; Xu et al., 2025; Zhang et al., 2025; Gao et al., 2025). This technique leverages the inherent sparsity of attention matrices, wherein most of the attention mass for a given query is concentrated on a small subset of key tokens. This property, particularly prominent in long sequences, allows for a drastic reduction in computation without significantly compromising performance. While this block-level approach maximizes parallel efficiency, its rigidity can lead to a sub-optimal sparsity pattern. Important key tokens (often referred to as "heavy hitters" (Zhang et al., 2023)) could typically be distributed in a long-tailed, scattered man-

ner across the sequence. This **information fragmentation** compels block-based methods to retrieve a large number of blocks to cover these sparse signals, resulting in a **diluted information distribution** and computational redundancy. Essentially, existing methods focus on *passively selecting* blocks from a chaotic matrix, rather than optimizing the matrix structure itself.

Fortunately, the same token-wise computation that leads to quadratic complexity also presents an opportunity to mitigate it. Since the attention mechanism is permutation-invariant, we can move beyond passive selection and actively **restructure** the query and key sequences to create a more favorable sparsity pattern. Leveraging this insight, we propose **Permuted Block-Sparse Attention (PBS-Attn)**, a plug-and-play strategy that reorganizes tokens to consolidate the attention mass for LLM prefilling. By clustering globally important tokens into contiguous regions, PBS-Attn transforms the scattered heavy hitters into high-density blocks, allowing the model to capture the majority of attention mass with significantly fewer block retrievals. To reconcile the conflict between permutation and the causal masking required by LLMs, we introduce a novel **Segmented Permutation** strategy that strictly preserves inter-segment causality while applying intra-segment permutation.

Extensive experiments demonstrate that PBS-Attn increases block-level sparsity, yielding significant efficiency gains with minimal degradation in model performance. Specifically, powered by our custom permuted-FlashAttention kernels, PBS-Attn achieves an end-to-end speedup of up to $2.75\times$ in LLM prefilling, while maintaining performance close to the full attention baseline on datasets like Long-Bench (Bai et al., 2024), LongBenchv2 (Bai et al., 2025) and RULER (Hsieh et al., 2024).

## 2. Motivation and Analysis

In this section, we analyze the limitations of existing block-sparse mechanisms through the lens of *information fragmentation* and demonstrate how token permutation serves as an effective strategy for **structural consolidation**. For brevity, we assume the standard formulation of attention where inputs $\mathbf{Q}, \mathbf{K}, \mathbf{V}$ are partitioned into blocks of size $B$. Detailed preliminaries and notations are provided in Appendix A.

### 2.1. The Problem of Information Fragmentation

Although the attention matrix is intrinsically sparse at the token level, block-sparse attention operates at a coarser granularity to leverage hardware efficiency. For a given query block indexed by $B$, let the set of theoretically critical key tokens be $\mathcal{K}_B$. This set is the union of individual top-$k$

key tokens $\mathcal{S}_i$ for each query $q_i \in B$:

$$\mathcal{K}_B = \bigcup_{i \in B} \mathcal{S}_i \tag{1}$$

The objective of block-sparse attention is to select a set of key blocks $\mathbb{C}_{\text{sel}}$ to maximize the captured attention mass subject to a budget constraint $m_\alpha$:

$$\max_{\mathbb{C}_{\text{sel}}} \sum_{C \in \mathbb{C}_{\text{sel}}} \sum_{i \in B} \sum_{j \in C} A_{i,j} \quad \text{s.t.} \quad |\mathbb{C}_{\text{sel}}| \leq m_\alpha \tag{2}$$

where $A_{i,j}$ is the oracle attention score.

The challenge arises when the tokens in $\mathcal{K}_B$ are scattered sparsely across the sequence, i.e., **information fragmentation**. In the worst-case scenario where the tokens are uniformly distributed, recovering $\mathcal{K}_B$ requires retrieving every block that contains even a single relevant token. This leads to substantial computational redundancy, as the retrieved blocks contain mostly irrelevant noise. Existing methods typically accept this structure as immutable and attempt to find the optimal $\mathbb{C}_{\text{sel}}$. We argue that optimizing $\mathbb{C}_{\text{sel}}$ on a fragmented structure yields diminishing returns. **Hence, we seek to optimize the attention matrix structure itself as a new axis of optimization.**

### 2.2. Permutation as Structural Consolidation

We hypothesize that by reordering the key sequence, we can cluster $\mathcal{K}_B$ into contiguous regions, thereby **consolidating** the attention mass. This structural optimization allows the model to capture the same attention mass with significantly fewer blocks ($|\mathbb{C}_{\text{sel}}|$).

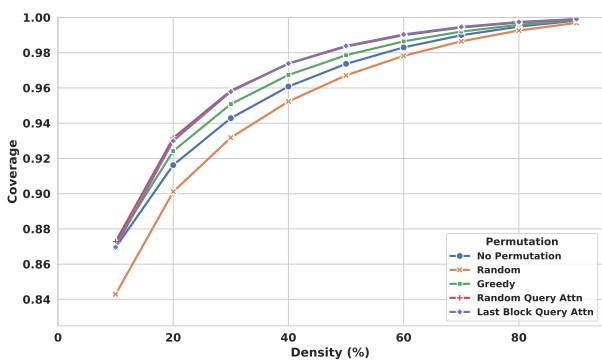

*Figure 1.* Coverage-density trade-off of various permutation strategies on Llama-3.1-8B-Instruct (16K context). Strategies that cluster globally important keys (Random/Last-Block Query-Attn) significantly outperform the baseline and local greedy alignment.

To validate this and determine the optimal permutation strategy, we evaluate four permutation heuristics on Llama-3.1-8B-Instruct (on 16K context length) by measuring their coverage-density trade-off (defined in Eq. 2): (1) **No Permutation:** The baseline natural ordering. (2) **Random**

**Permutation:** Randomly shuffles tokens to test if permutation alone improves coverage without guidance. (3) **Greedy Query-Aware Key Permutation:** A strategy representing *fine-grained local alignment*. For each query block, we compute its centroid (via mean pooling) and iteratively assign the most similar available key tokens (based on cosine similarity). This explicitly attempts to match keys to specific local query contexts. (4) **Global-Importance-based Key Permutation:** Sorts keys based on their accumulated attention scores calculated from a subset of queries (either a random subset or the last block).

**Key Insight: Global Clustering $\succ$ Local Alignment.** Figure 1 presents the results, revealing four critical insights:

- **Long-Tailed Distribution & Diminishing Returns:** The coverage-density distribution follows a long-tailed pattern, implying diminishing returns for retrieval in the natural order. Permutation effectively "compresses" this tail, shifting meaningful information into high-density blocks.

- **The Original Locality:** Random permutation significantly degrades coverage, indicating that the natural order contains some local structure. However, it is far from optimal.

- **Local Alignment Helps but Global "Heavy Hitters" Dominate:** The strategies based on Global Importance (both "Random-Query" and "Last-Block-Query") achieve the highest coverage, significantly outperforming the local Greedy approach. This suggests that the primary gain from permutation comes from **clustering globally critical "heavy hitter" tokens** (Zhang et al., 2023) rather than fine-grained query-key alignment.

- **Robustness of Proxy Scoring:** The performance gap between using a random subset of queries versus the last block of queries is negligible. This implies that the set of heavy hitters can be estimated using any subset of queries.

This analysis provides the theoretical foundation for our proposed method in Section 3, justifying the use of a lightweight, proxy-based sorting mechanism. To further understand the boundary of efficacy for permutation, we dissect the sparsity gains across different layers and heads, including a failure mode analysis for specific patterns. Across all 1024 heads of Llama-3.1-8B at 32K context and 97.5% coverage, permutation improves block-level sparsity for 70.8% of heads and harms only 5.2%. Detailed analyses are provided in Appendix B.

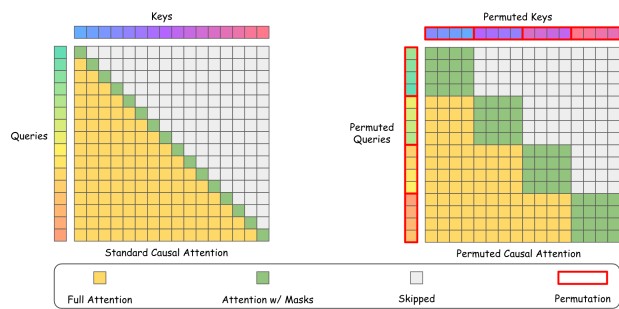

*Figure 2.* Illustration of causal attention without (**Left**) and with (**Right**) segmented permutation with $B = 1, S = 4$. Segmented permutation enhances block-level sparsity via intra-segment permutation while preserving inter-segment causality. By restricting computation of blocks within on-diagonal segments (green blocks), we can safely skip inter-segment blocks (yellow blocks) for block-sparse attention.

## 3. Permuted Block-Sparse Attention

Building on the insight from Section 2, in this work, we propose **Permuted Block-Sparse Attention (PBS-Attn)**, a novel approach that optimizes block-level sparsity by leveraging the permutation properties of attention.

### 3.1. Theoretical Foundation: Permutation Properties of Attention

The attention mechanism exhibits specific symmetries with respect to permutations of its inputs, which we formalize in the following lemmas.

**Lemma 3.1** (Key-Value Pair Permutation Invariance). *The attention mechanism is invariant to the order of the source sequence, provided that the key-value pairings are maintained.*

*Formally, let $\mathbf{P}_\pi \in \{0,1\}^{M \times M}$ be a permutation matrix that reorders the rows of a matrix according to a permutation $\pi$ on the index set $\{1, \ldots, M\}$. The following identity holds:*

$$Attention(\mathbf{Q}, \mathbf{P}_\pi \mathbf{K}, \mathbf{P}_\pi \mathbf{V}) = Attention(\mathbf{Q}, \mathbf{K}, \mathbf{V}) \quad (3)$$

**Lemma 3.2** (Query Permutation Equivariance). *The attention mechanism is equivariant with respect to permutations of the query sequence.*

*Formally, let $\mathbf{P}_\sigma \in \{0,1\}^{N \times N}$ be a permutation matrix that reorders the rows of a matrix according to a permutation $\sigma$ on the index set $\{1, \ldots, N\}$. The following relationship holds:*

$$Attention(\mathbf{P}_\sigma \mathbf{Q}, \mathbf{K}, \mathbf{V}) = \mathbf{P}_\sigma Attention(\mathbf{Q}, \mathbf{K}, \mathbf{V}) \quad (4)$$

The proofs of Lemma 3.1 and 3.2 are provided in Appendix C.1 and C.2, respectively.

Combining these properties, we arrive at a general theorem for attention under simultaneous input permutations. A detailed proof is provided in Appendix C.3.

**Theorem 3.3** (Attention Permutation Invariance under Inverse Transformation). *If the queries are permuted by $\mathbf{P}_\sigma$ and the key-value pairs are permuted by $\mathbf{P}_\pi$, the resulting output is a permuted version of the original output. Applying the inverse of the query permutation recovers the original, unpermuted output. Formally:*

$$\mathbf{P}_\sigma^T \, Attention(\mathbf{P}_\sigma\mathbf{Q}, \mathbf{P}_\pi\mathbf{K}, \mathbf{P}_\pi\mathbf{V}) = Attention(\mathbf{Q}, \mathbf{K}, \mathbf{V}) \tag{5}$$

Theorem 3.3 establishes that the query matrix $\mathbf{Q}$ and key matrix $\mathbf{K}$ can be permuted by $\mathbf{P}_\sigma$ and $\mathbf{P}_\pi$ respectively, provided that $\mathbf{P}_\pi$ is also applied to the value matrix $\mathbf{V}$ and $\mathbf{P}_\sigma^T$ to the output $\mathbf{O}'$. This property enables the rearrangement of the attention matrix $\mathbf{A}$, without affecting the attention output.

### 3.2. Segmented Permutation for Causal Attention

While Theorem 3.3 provides the theoretical basis for rearrangement, a critical challenge remains: **maintaining causality post-permutation**. Specifically, LLMs are trained with causal attention, which restricts queries to attending only to keys in preceding positions, resulting in a lower-triangular attention matrix, $\mathbf{A}$. During prefilling, blocks above the main diagonal are computationally redundant and can be skipped; consequently, the original block density for causal attention is $\frac{T_c+1}{2T_c}$. A naive application of a global permutation to the query and key sequences would dismantle this vital causal structure. Such a permutation could scatter dependencies across the entire matrix, potentially transforming the sparse, lower-triangular structure into a fully dense one (i.e., a block density of 1).

To address this challenge, we propose a segmented permutation strategy that preserves *inter-segment causality* while applying *intra-segment permutation*, illustrated in Figure 2.

Formally, we partition the initial $\lfloor N/S \rfloor \cdot S$ tokens of the input sequences $\mathbf{Q}, \mathbf{K}, \mathbf{V}$ into $G = \lfloor N/S \rfloor$ non-overlapping, contiguous segments of size $S$. The remaining $N \pmod{S}$ tokens are left unpermuted.

Let $\mathbf{Q}_i, \mathbf{K}_i, \mathbf{V}_i \in \mathbb{R}^{S \times d}$ denote the $i$-th segment for $i \in \{1, \dots, G\}$. For each segment $i$, we introduce local permutations, $\sigma_i$ for queries and $\pi_i$ for keys, that reorder tokens within that segment. The global permutation operators, $\mathbf{P}_\sigma$ and $\mathbf{P}_\pi$, are then constructed as block-diagonal matrices from these respective local permutations. For the key per-

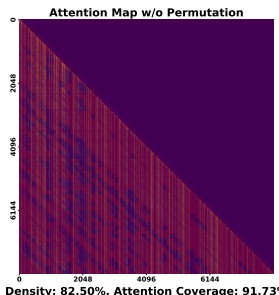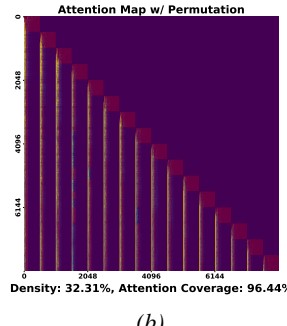

*Figure 3.* Comparison of attention maps for Llama-3.1-8B on an 8K LongBench example, showing the pattern without (a) and with (b) segmented permutation. The red overlay indicates blocks selected for block-sparse attention, and the attention coverage is calculated as the total attention scores covered by the selected blocks. More visualizations are provided in Appendix H.

mutation matrix $\mathbf{P}_\pi$:

$$\mathbf{P}_\pi = \mathrm{diag}(\mathbf{P}_{\pi_1}, \dots, \mathbf{P}_{\pi_G}, \mathbf{I}_{N \pmod{S}})$$
$$= \begin{pmatrix} \mathbf{P}_{\pi_1} & \mathbf{0} & \cdots & & \mathbf{0} \\ \mathbf{0} & \mathbf{P}_{\pi_2} & \cdots & & \mathbf{0} \\ \vdots & \vdots & \ddots & & \vdots \\ \mathbf{0} & \mathbf{0} & \cdots & \mathbf{I}_{N \pmod{S}} \end{pmatrix} \tag{6}$$

Here, each $\mathbf{P}_{\pi_i} \in \{0, 1\}^{S \times S}$ is the permutation matrix for the local key permutation $\pi_i$, and $\mathbf{I}_{N \pmod{S}}$ is the identity matrix corresponding to the last incomplete segment. The query permutation matrix $\mathbf{P}_\sigma$ is constructed analogously from its own set of local permutations, $\{\sigma_i\}_{i=1}^G$.

### 3.3. Global-Importance-based Permutation

The core of PBS-Attn is determining the optimal local permutation $\pi_i$ for each segment to maximize block-level sparsity. Based on the key insight from Section 2.2, where global heavy hitters dominate and their importance is consistent across queries, we sort the keys based on their global importance proxied by a small subset of queries.

Concretely, we utilize the queries from the last block, $\mathbf{Q}_{\text{last\_block}}$, to estimate the global importance of all keys. We compute a global importance score vector $\mathbf{s} \in \mathbb{R}^N$:

$$\mathbf{s} = \mathrm{mean}_{\text{rows}} \left( \mathrm{softmax} \left( \frac{\mathbf{Q}_{\text{last\_block}}\mathbf{K}^T}{\sqrt{d}} \right) \right) \tag{7}$$

Computing this proxy score involves a matrix multiplication of size $(B \times d)$ and $(N \times d)$, resulting in a linear complexity of $O(N \cdot B \cdot d)$, which is negligible in long-context prefilling. The local permutation $\pi_i$ for each segment $i$ is then obtained by sorting the keys within that segment based on $\mathbf{s}$ in descending order:

$$\pi_i = \mathrm{argsort}(-\mathbf{s}_{[(i-1)S+1:iS]}) \tag{8}$$

---

**Algorithm 1** Permuted Block-Sparse Attention

---

**Require:** $\mathbf{Q}, \mathbf{K}, \mathbf{V} \in \mathbb{R}^{N \times d}$, permutation matrices $\mathbf{P}_\sigma, \mathbf{P}_\pi \in \{0,1\}^{N \times N}$, segment size $S$, block size $B$

**Ensure:** Permuted attention output $\mathbf{O} \in \mathbb{R}^{N \times d}$

  $\mathbf{Q}' \leftarrow \mathbf{P}_\sigma \mathbf{Q}, \mathbf{K}' \leftarrow \mathbf{P}_\pi \mathbf{K}, \mathbf{V}' \leftarrow \mathbf{P}_\pi \mathbf{V}$ {Apply permutation}

  Divide $\mathbf{Q}'$ into $T_r = \lceil \frac{N}{B} \rceil$ blocks $\mathbf{Q}'_1, \ldots, \mathbf{Q}'_{T_r}$; divide $\mathbf{K}', \mathbf{V}'$ into $T_c = \lceil \frac{N}{B} \rceil$ blocks $\mathbf{K}'_1, \ldots, \mathbf{K}'_{T_c}$ and $\mathbf{V}'_1, \ldots, \mathbf{V}'_{T_c}$;

  $\mathbf{M} \leftarrow \text{BLOCK\_SELECTION}(\mathbf{Q}', \mathbf{K}', B, S)$ {Appendix D}

  Initialize $\mathbf{O}' \leftarrow \mathbf{0}$;

  **for** $i = 1$ to $T_r$ **do**

    Load $\mathbf{Q}'_i$ to SRAM; Initialize $\mathbf{O}^{(0)}_i \leftarrow \mathbf{0}, \mathbf{m}^{(0)}_i \leftarrow -\infty, \mathbf{l}^{(0)}_i \leftarrow \mathbf{0}$;

    **for** $j = 1$ to $T_c$ **do**

      **if** $\mathbf{M}_{i,j} = 1$ **then** {Compute attention only for selected blocks}

        Load $\mathbf{K}'_j, \mathbf{V}'_j$ to SRAM;

        $\mathbf{S}'_{ij} = \mathbf{Q}'_i \mathbf{K}'^T_j / \sqrt{d}, \mathbf{m}^{(j)}_i = \max(\mathbf{m}^{(j-1)}_i, \text{row\_max}(\mathbf{S}'_{ij}))$;

        $\mathbf{l}^{(j)}_i = \mathbf{l}^{(j-1)}_i e^{\mathbf{m}^{(j-1)}_i - \mathbf{m}^{(j)}_i} + \text{row\_sum}(\exp(\mathbf{S}'_{ij} - \mathbf{m}^{(j)}_i))$;

        $\mathbf{O}^{(j)}_i = \mathbf{O}^{(j-1)}_i e^{\mathbf{m}^{(j-1)}_i - \mathbf{m}^{(j)}_i} + \exp(\mathbf{S}'_{ij} - \mathbf{m}^{(j)}_i) \mathbf{V}'_j$;

      **else**

        $\mathbf{O}^{(j)}_i \leftarrow \mathbf{O}^{(j-1)}_i, \mathbf{m}^{(j)}_i \leftarrow \mathbf{m}^{(j-1)}_i, \mathbf{l}^{(j)}_i \leftarrow \mathbf{l}^{(j-1)}_i$; {Skip}

      **end if**

    **end for**

    $\mathbf{O}'_i \leftarrow \text{diag}((\mathbf{l}^{(T_c)}_i)^{-1}) \mathbf{O}^{(T_c)}_i$; Write $\mathbf{O}'_i$ back to its rows in $\mathbf{O}'$;

  **end for**

  $\mathbf{O} \leftarrow \mathbf{P}_\sigma^T \mathbf{O}'$ {Reverse permutation}

  **return** $\mathbf{O}$

---

This strategy aligns with our findings in Figure 1: it effectively clusters the "Vertical Lines" (global heavy hitters) into the leading blocks of each segment. As visualized in Figure 3b, this transformation significantly consolidates the attention mass, allowing high coverage with fewer blocks.

Regarding the query permutation $\mathbf{P}_\sigma$, our analysis in Figure 6a indicated marginal gains from permuting queries. Thus, we maintain the natural order of queries (i.e., $\mathbf{P}_\sigma = \mathbf{I}$) to preserve local context and minimize overhead.

### 3.4. Permuted Block-Sparse Attention

The proposed permuted block-sparse attention (**PBS-Attn**) mechanism is detailed in Algorithm 1, where the key adjustments relative to FlashAttention are highlighted in red. The process commences by permuting the query, key, and value matrices. Specifically, we apply permutation $P_\sigma$ to the query matrix $\mathbf{Q}$ and $P_\pi$ to the key matrix $\mathbf{K}$, while the value matrix $\mathbf{V}$ shares the same permutation $P_\pi$, as justified by Lemma 3.1. Subsequently, a block selection algorithm is applied to the permuted queries and keys, yielding a block-sparse mask $\mathbf{M}$. This mask, $\mathbf{M}$, governs the tiled attention computation by dictating which block-wise operations can be pruned. For selected blocks (where $\mathbf{M}_{i,j} = 1$), a standard online softmax attention is computed, updating the state of the permuted output block $\mathbf{O}'_i$. For unselected blocks (where $\mathbf{M}_{i,j} = 0$), this computation is skipped, and the state of $\mathbf{O}'_i$ remains unchanged. For the block selection algorithm, we use a simple strategy that utilizes mean pooling and block-wise attention to estimate the importance of each key block for each query block for the main method, where we detail in Appendix D.1. Crucially, we demonstrate that the sparsity improvements conferred by permutation

are agnostic to the specific block selection algorithm, allowing our method to be combined with existing algorithms to further improve block sparsity, as shown in Appendix D.2. Finally, an inverse permutation, $P_\sigma^T$, is applied to the output $\mathbf{O}'$ to restore the original ordering, as established by Theorem 3.3.

## 4. Experiments

### 4.1. Settings

**Models & Datasets** For the main experiments, we employ two state-of-the-art long-context LLMs, claiming support for available context lengths above 128K tokens: **Llama-3.1-8B(128K)** (Grattafiori et al., 2024) and **Qwen-2.5-7B-1M(1M)** (Yang et al., 2025a). We evaluate the sparse attention methods on two challenging real-world long-context datasets to validate their effectiveness in real-world scenarios: **LongBench** (Bai et al., 2024) and **LongBenchv2** (Bai et al., 2025). LongBench is a collection of 21 long-context understanding tasks in 6 categories with mostly real-world data, with the average length of most tasks ranging from 5K to 15K. LongBenchv2 further scales the context length, ranging from 8K to 2M, covering various realistic scenarios. We also conduct evaluation on **RULER** (Hsieh et al., 2024), a synthetic benchmark designed to systematically evaluate long-context LLMs across various context lengths.

**Baselines** We evaluate PBS-Attn alongside a set of strong baselines to validate its effectiveness. (1) **Full Attention**: The standard attention mechanism that computes the full attention matrix as the oracle method. Specifically, we use the FlashAttention (Dao et al., 2022) implementation. (2) **Minference** (Jiang et al., 2024): A sparse attention method that performs offline attention pattern search, we utilize the official configuration for attention pattern setting. (3) **FlexPrefill** (Lai et al., 2025): A block selection method for block-sparse attention that performs block selection based on the input and selects the attention pattern on the fly. We use $\gamma = 0.95, \tau = 0.1$ as reported in the original paper. (4) **XAttention** (Xu et al., 2025): A block selection method for block-sparse attention that selects blocks based on an antidiagonal scoring of blocks. We use threshold $= 0.9$, stride $= 8$ as reported in the original paper. (5) **MeanPooling**: This method uses a mean pooling strategy on the unpermuted queries and keys to select blocks, which is the same selection method for PBS-Attn(detailed in D.1). Our experiments shows that MeanPooling can serve as a strong baseline when the first and the most recent key blocks are forcibly selected for each query block, due to the attention sink phenomenon (Xiao et al., 2024). We use a selection threshold of 0.9 for MeanPooling.

*Table 1.* Performance comparison of various sparse attention methods on LongBench. **Bold** and underlined scores indicate the best and second-best performing methods in each category, respectively, with the exception of the full attention baseline.

| Method | Single-Doc QA | Multi-Doc QA | Summarization | Few-shot Learning | Code | Synthetic | Avg. |
|---|---|---|---|---|---|---|---|
| | | | *Llama-3.1-8B* | | | | |
| Full | 48.80 | 41.80 | 17.79 | 29.73 | 24.77 | 66.82 | 38.28 |
| MInference | 47.21 | 40.93 | 17.72 | 29.36 | 24.77 | 62.36 | 37.06 |
| FlexPrefill | 47.03 | 38.57 | 17.78 | 30.38 | 24.88 | 24.71 | 30.56 |
| XAttention | **48.26** | 40.23 | **17.86** | **31.35** | **26.19** | 54.64 | 36.42 |
| MeanPooling | 46.61 | 40.66 | 17.85 | 30.64 | 26.10 | 58.14 | 36.67 |
| **PBS-Attn** | 48.00 | **42.09** | 17.72 | 28.36 | 24.25 | **63.80** | **37.37** |
| | | | *Qwen-2.5-7B-1M* | | | | |
| Full | 44.21 | 42.97 | 16.01 | 47.48 | 3.91 | 67.50 | 37.01 |
| MInference | 42.82 | 41.76 | 16.01 | 46.41 | 3.80 | **66.50** | 36.21 |
| FlexPrefill | 38.44 | 37.51 | 15.87 | 46.12 | **6.46** | 26.67 | 28.51 |
| XAttention | **43.82** | **42.21** | 16.07 | 48.30 | 3.83 | 63.33 | 36.26 |
| MeanPooling | 39.39 | 40.96 | 15.95 | **49.07** | 4.80 | 40.83 | 31.83 |
| **PBS-Attn** | 43.01 | 41.40 | **16.12** | 47.36 | 4.00 | 66.33 | **36.37** |

**Implementation Details** For PBS-Attn, we use a block size of $B = 128$ and a segment size of $S = 256$. The block selection threshold is set to $0.9$ through all experiments. We implement a custom permuted-FlashAttention kernel in Triton (Tillet et al., 2019) for efficient inference of PBS-Attn. To handle Grouped Query Attention (GQA), our default strategy replicates keys and values within each group to maximize sparsity gains. We also evaluate the feasibility of sharing the permutation within a GQA group to improve memory efficiency, as detailed in Appendix G. The experiments are conducted in a computing environment with NVIDIA H100 80GB GPUs.

*Table 2.* Performance comparison of various sparse attention methods on LongBenchv2. **Bold** and underlined scores indicate the best and second-best performing methods for each model, respectively, with the exception of the full attention baseline.

| Method | Llama-3.1-8B | Qwen2.5-7B-1M |
|---|---|---|
| Full | 28.83 | 35.19 |
| Minference | 29.03 | 34.19 |
| FlexPrefill | 27.24 | 27.83 |
| XAttention | 29.62 | 34.19 |
| MeanPooling | 29.42 | 26.24 |
| **PBS-Attn** | **29.82** | **34.39** |

### 4.2. Main Results

**LongBench** Table 1 presents a performance comparison of various sparse attention methods on the LongBench benchmark, evaluated using the Llama-3.1-8B-Instruct and Qwen-2.5-7B-1M models. As the results indicate, the unpermuted MeanPooling method already establishes a strong baseline. Crucially, by incorporating our proposed permutation strategy, PBS-Attn significantly improves performance, surpassing other block-sparse attention methods and closely approaching the performance of the oracle full-attention baseline. PBS-Attn consistently achieves the best overall performance across both models, demonstrating its effec-

tiveness and robustness. To further demonstrate model generalization, Appendix F reports additional evaluations on Qwen3-8B and Qwen-2.5-14B-Instruct-1M. On Qwen3-8B, PBS-Attn matches full attention on LongBench (33.98 vs. 34.08 average score) and achieves up to a $2.72\times$ end-to-end speedup.

**LongBenchv2** To rigorously evaluate the effectiveness of PBS-Attn in extreme long-context scenarios, we conducted experiments on the more challenging LongBenchv2 benchmark. The results, presented in Table 2, reveal that PBS-Attn exhibits minimal performance degradation compared to the full attention baseline while consistently surpassing other block-sparse attention methods. Notably, PBS-Attn consistently outperforms the unpermuted MeanPooling baseline. This advantage is particularly pronounced for the Qwen-2.5-7B-1M model, where permutation brings a remarkable relative improvement of $31\%$ in overall performance. This indicates that permutation can successfully group key tokens that have similar behaviors, making the block selection more precise and covering more critical key tokens.

**RULER** To systematically evaluate PBS-Attn across various lengths, we conduct experiments on the RULER benchmark, with results presented in Table 3. Due to the synthetic nature of the RULER dataset, mean pooling selection drastically diminishes performance on tasks retrieving key-value pairs in random UUIDs, necessitating the use of token-level attention in block scoring for these tasks. Therefore, we also incorporate PBS-Attn[+], which adopts the antidiagonal block scoring scheme proposed in XAttention (Xu et al., 2025). Notably, both PBS-Attn and PBS-Attn[+] consistently outperform their unpermuted baselines, MeanPooling and XAttention, respectively, demonstrating the effectiveness of permutation. Concretely, PBS-Attn achieves an average score improvement of 3.21 over MeanPooling on Llama-3.1-8B-Instruct; this gain is particularly pronounced at longer

*Table 3.* Results on the RULER benchmark. PBS-Attn[+] denotes PBS-Attn with antidiagonal scoring for block selection (Xu et al., 2025).

| Method | Llama-3.1-8B | | | | | | | Qwen-2.5-7B-1M | | | | | | |
|---|---|---|---|---|---|---|---|---|---|---|---|---|---|---|
| | 4K | 8K | 16K | 32K | 64K | 128K | Avg | 4K | 8K | 16K | 32K | 64K | 128K | Avg |
| Full | 96.14 | 94.24 | 92.19 | 86.06 | 84.60 | 75.30 | 88.09 | 95.34 | 92.45 | 93.49 | 89.06 | 84.73 | 74.23 | 88.22 |
| Minference | **95.98** | 93.67 | **91.95** | 85.55 | **83.48** | 70.47 | 86.85 | 94.01 | 91.30 | 91.60 | 89.09 | 81.30 | 70.10 | 86.23 |
| FlexPrefill | 92.87 | 92.99 | 91.35 | 84.91 | 82.62 | 71.07 | 85.97 | 84.17 | 87.74 | 86.73 | 84.21 | 78.15 | 61.66 | 80.44 |
| XAttention | 95.63 | **93.95** | 91.63 | 86.32 | 80.54 | 70.68 | 86.46 | 93.69 | 92.10 | 91.50 | 88.35 | 81.26 | 73.05 | 86.66 |
| MeanPooling | 94.15 | 92.72 | 89.94 | 83.95 | 76.46 | 59.32 | 82.76 | 90.15 | 87.43 | 86.38 | 82.70 | 78.86 | 67.51 | 82.17 |
| **PBS-Attn** | 95.83 | 93.85 | 91.46 | 85.18 | 82.51 | 66.98 | 85.97 | 93.27 | 90.77 | 90.54 | 85.54 | 81.50 | 70.61 | 85.37 |
| **PBS-Attn[+]** | 95.72 | 93.85 | 91.23 | **87.05** | 81.27 | **72.09** | **86.87** | **94.06** | **92.24** | **92.59** | **89.31** | **84.37** | **73.71** | **87.71** |

contexts, reaching an improvement of 7.66 at 128K. PBS-Attn[+] further enhances performance, exceeding XAttention by 1.41 on Llama-3.1-8B-Instruct and 1.05 on Qwen-2.5-7B-1M, approaching the full attention baselines with narrow margins of 3.21 and 0.51, respectively.

**Efficiency Results** To best evaluate the real-world practicality of the sparse attention methods, we measure the end-to-end time to first token (TTFT) on sequence lengths ranging from 8K to 512K. As shown in Figure 4, PBS-Attn achieves the highest speedup across all context lengths, whereas most competing methods only excel within a limited range. For instance, Minference does not show a speedup over FlashAttention until 128k, and the efficiency gains of XAttention stagnate after 128K. Although Flex-Prefill matches the speedup of PBS-Attn in most cases, it suffers from a significant quality drop as shown in Table 1 and 2. In contrast, PBS-Attn consistently delivers the best performance, reaching a $2.75\times$ end-to-end speedup at 256K, demonstrating its superior practicality and robustness. To analyze the permutation overhead in PBS-Attn, we further conduct a detailed benchmarking study in Appendix E.

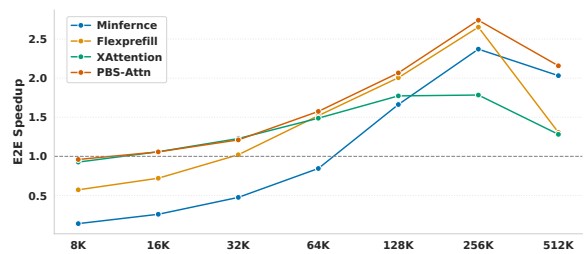

*Figure 4.* Speedup of various methods relative to FlashAttention, measured by time to first token (TTFT) on LongBenchv2 across various sequence lengths. To accommodate longer sequences under memory constraints, we employ tensor parallelism with tp_size of 2 and 8 for the 256K and 512K contexts, respectively.

### 4.3. Ablation Studies and Analysis

**Impact of Permutation on Sparsity.** Figure 5 quantifies the structural gain. Permutation consistently lifts the sparsity curve, achieving a 7% absolute improvement at just 8K

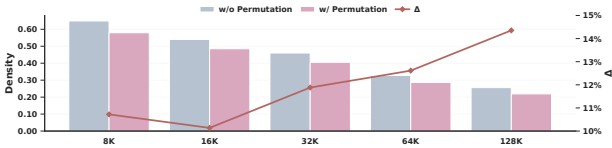

*Figure 5.* Block-level density on various context lengths with and without permutation. A relative sparsity improvement $\Delta$ is calculated.

context. This gap widens as sequence length increases, validating that structural optimization becomes more critical for longer, more fragmented contexts. Appendix B.2 further converts these density results into selected-block counts, showing that permutation reduces selected blocks by 10.7% at 8K and 14.4% at 128K.

**Design Choice: Why Permute Keys but not Queries?** To analyze the effect of permutation on queries, we propose a key-aware query permutation approach. However, the attention distribution of queries over keys is often less structured than that of keys over queries. We therefore employ a straightforward strategy that clusters queries which attend to similar keys within a given segment. Specifically, we first compute a set of centroids by calculating block-averaged keys, denoted as $\bar{\mathbf{K}}$. Each centroid is defined as $\bar{\mathbf{K}}_i = \text{MeanPool}(\mathbf{K}_{[(i-1)B+1:iB]})$ for $i = 1, \ldots, T_c$. We then determine cluster assignments by computing the cosine similarity between each query and these centroids. Within each segment, queries are assigned greedily based on their similarity to the centroids. We evaluate the effect of the permutation target and order in Figure 6a. The results indicate that permuting both queries and keys brings no noticeable improvements, regardless of the order. Permuting queries offers a marginal improvement over permuting keys in the performance-density trade-off, but it can be less efficient considering Grouped-Query Attention (GQA) (Ainslie et al., 2023). Accordingly, we exclusively adopt query-aware key permutation in our main method.

**Effect of Segment Size** Segment size $S$ plays a crucial role in segmented permutation, where tokens are per-

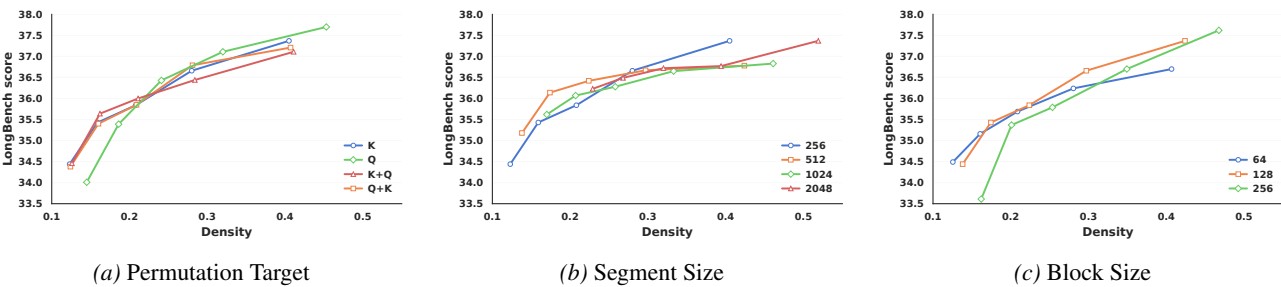

*(a)* Permutation Target        *(b)* Segment Size        *(c)* Block Size

*Figure 6.* LongBench score vs. average block-level density at a context length of 32K.

muted within the corresponding segments to maintain inter-segment causality. Intuitively, a larger segment size $S$ takes into account more tokens during sorting, thereby enhancing block-level sparsity; however, it would also include more blocks in the on-diagonal segments, which can not be skipped during computation to avoid breaking causality. Figure 6b illustrates how the segment size, $S$, affects the performance-density trade-off. A larger $S$ flattens the trade-off curve, indicating that segmented permutation effectively clusters key tokens, allowing the model to maintain high performance even at high levels of block-level sparsity. However, this benefit diminishes at lower sparsity levels, as the wide on-diagonal segments contain a large number of blocks that must be computed, limiting block-level sparsity.

**Effect of Block Size**    We analyze the impact of block size $B$ on the performance-density trade-off in Figure 6c. Smaller blocks ($B = 64$) provide finer granularity, yielding better performance at very low densities ($< 0.15$) by minimizing redundancy. However, larger blocks ($B = 256$) suffer from rapid degradation at low budgets, as coarse selection forces the inclusion of non-critical tokens. The intermediate size ($B = 128$) strikes the optimal balance, achieving the highest LongBench scores across most density levels while maintaining robustness. We therefore select $B = 128$ for our main experiments.

## 5. Related Work

**Sparse Attention**    The quadratic growth in memory and computational requirements of the attention mechanism has been a bottleneck for scaling LLM context lengths. Sparse attention has emerged as a promising solution, leveraging the inherent sparsity in attention patterns to drastically reduce this overhead. StreamingLLM (Xiao et al., 2024) first identifies the attention sink phenomenon in LLMs, proposing to capture a majority of the attention mass with initial and recent tokens. NSA (Yuan et al., 2025) and MoBA (Lu et al., 2025) further incorporate sparse attention into the training stage, accelerating both prefilling and decoding. Methods like H2O (Zhang et al., 2023), can accelerate the decoding speed by exploiting the attention pattern after

prefilling. Closely related to this work, various methods are proposed to accelerate the compute-bounded prefilling process. For example, Minference (Jiang et al., 2024) recognizes attention patterns in a pre-computed manner. More recent works tend to perform attention pattern recognition on-the-fly. For instance, FlexPrefill (Lai et al., 2025) utilizes divergence to classify the attention pattern, XAttention (Xu et al., 2025) adopts an antidiagonal scoring metric to weight each block, SpargeAttention (Zhang et al., 2025) accounts the intra-block similarity into the selection criterion, and Prism (Wang et al., 2026) utilizes a dual-band importance estimation for block selection. However, these methods primarily focus on developing better block selection algorithms, while our work is orthogonal: we focus on rearranging the attention matrix to create a structure that inherently increases block-level sparsity.

**Attention with Token Permutation**    The idea of reordering tokens to optimize attention computation was pioneered by Reformer (Kitaev et al., 2020), which employs Locality-Sensitive Hashing (LSH) to bucket similar queries and keys, thereby reducing attention computation complexity. However, Reformer relies on a Shared-QK formulation, making it incompatible with modern pre-trained LLMs without significant architectural changes and retraining. In contrast, PBS-Attn is designed as a plug-and-play method and can be applied to any modern LLM without additional training. Concurrent to our work, MMInference (Li et al., 2025) applies modality-aware permutation to accelerate long-context VLM prefilling, with permutations driven by modality structure rather than data-dependent importance, making it inapplicable to text-only LLMs. SVG2 (Yang et al., 2025b) and PAROAttention (Zhao et al., 2025) show promise in accelerating visual generation models like Diffusion Transformers (Peebles & Xie, 2023), but their reliance on bidirectional attention makes them incompatible with the causal constraints of auto-regressive LLMs. PBS-Attn addresses this causality challenge by introducing a segmented permutation strategy, explicitly preserving inter-segment causality.

# 6. Conclusion

In this work, we formalize the permutation properties of the attention mechanism and leverage them to improve block-level sparsity. We introduce Permuted Block-Sparse Attention (PBS-Attn), a plug-and-play method that employs a novel segmented permutation strategy to preserve inter-segment causality while reordering tokens within each segment. Our method achieves an end-to-end prefilling speedup of up to $2.75\times$ with minimal performance degradation, demonstrating a promising path toward more efficient long-context LLMs.

# Impact Statement

This paper focuses on improving the computational efficiency of long-context Large Language Models (LLMs) via block-sparse attention. By reducing the computational overhead and memory requirements of prefilling, our work contributes to lowering the energy consumption and carbon footprint associated with running large-scale models.

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

# A. Preliminaries

In this section, we provide the foundational mathematical formulations for standard attention, FlashAttention, and block-sparse attention, which serve as the basis for our proposed method.

**Scaled Dot-Product Attention**  As the cornerstone of modern large language models, the attention mechanism facilitates a dynamic synthesis of information by calculating a weighted aggregation of value ($\mathbf{V}$) vectors. These weights, or attention scores, are determined by the dot-product similarity between a given token's query ($\mathbf{Q}$) vector and the key ($\mathbf{K}$) vectors of all other tokens in the sequence. This process allows the model to directly assess the relevance of every token relative to every other, enabling the effective capture of long-range dependencies, but at a cost of quadratic complexity over the sequence length. Formally, the attention mechanism is given by:

$$\mathbf{A} = \text{softmax}\left(\frac{\mathbf{Q}\mathbf{K}^T}{\sqrt{d}}\right) \tag{9}$$

$$\text{Attention}(\mathbf{Q}, \mathbf{K}, \mathbf{V}) = \mathbf{A}\mathbf{V} \tag{10}$$

where $d$ is the head dimension for multi-head attention and $\mathbf{A}$ is the attention matrix.

**FlashAttention**  FlashAttention (Dao et al., 2022) employs a tiled approach that partitions the input sequence into blocks and performs an online softmax computation. This strategy circumvents the materialization of the full attention matrix $\mathbf{A}$, which significantly reduces memory overhead and improves efficiency for I/O-bound operations on GPUs.

Formally, let the input query, key and value matrices be $\mathbf{Q} \in \mathbb{R}^{N \times d}$, $\mathbf{K} \in \mathbb{R}^{M \times d}$, and $\mathbf{V} \in \mathbb{R}^{M \times d}$ and divide them into $T_r = \lceil \frac{N}{B} \rceil$ and $T_c = \lceil \frac{M}{B} \rceil$ blocks with block size $B$ (we use the same block size for $\mathbf{Q}$ and $\mathbf{K/V}$ for simple terminology), $\mathbf{Q} = [\mathbf{Q}_1, \ldots, \mathbf{Q}_{T_r}]$, $\mathbf{K} = [\mathbf{K}_1, \ldots, \mathbf{K}_{T_c}]$, and $\mathbf{V} = [\mathbf{V}_1, \ldots, \mathbf{V}_{T_c}]$. For query block $\mathbf{Q}_i$, the computation for the corresponding output block $\mathbf{O}_i$ is defined by a system of recursive equations over the key/value blocks $j = 1, \ldots, T_c$. The state at step $j$ is the triplet $(\mathbf{O}_i^{(j)}, \mathbf{m}_i^{(j)}, \mathbf{l}_i^{(j)})$. The state is initialized at $j = 0$ with $\mathbf{O}_i^{(0)} = \mathbf{0}$, $\mathbf{m}_i^{(0)} = -\infty$, and $\mathbf{l}_i^{(0)} = \mathbf{0}$. For each step $j = 1, \ldots, T_c$, given the intermediate scores $\mathbf{S}_{ij} = \frac{\mathbf{Q}_i \mathbf{K}_j^T}{\sqrt{d}}$ and local maximum $\mathbf{m}'_{ij} = \text{row\_max}(\mathbf{S}_{ij})$, the state is updated from $j - 1$ to $j$:

$$\mathbf{m}_i^{(j)} = \max(\mathbf{m}_i^{(j-1)}, \mathbf{m}'_{ij}) \tag{11}$$

$$\mathbf{l}_i^{(j)} = \mathbf{l}_i^{(j-1)} e^{\mathbf{m}_i^{(j-1)} - \mathbf{m}_i^{(j)}} + \text{row\_sum}(\exp(\mathbf{S}_{ij} - \mathbf{m}_i^{(j)})) \tag{12}$$

$$\mathbf{O}_i^{(j)} = \mathbf{O}_i^{(j-1)} e^{\mathbf{m}_i^{(j-1)} - \mathbf{m}_i^{(j)}} + \exp(\mathbf{S}_{ij} - \mathbf{m}_i^{(j)})\mathbf{V}_j \tag{13}$$

After the final step, the output is normalized as $\mathbf{O}_i = \text{diag}\left((\mathbf{l}_i^{(T_c)})^{-1}\right)\mathbf{O}_i^{(T_c)}$.

**Block-Sparse Attention**  Building upon the tiled computation of FlashAttention, block-sparse attention introduces a further layer of optimization by selectively pruning block-wise interactions. This is achieved using a predefined sparse block mask, $\mathbf{M} \in \{0, 1\}^{T_r \times T_c}$. For any given query block $\mathbf{Q}_i$, the attention computation is only performed against key-value blocks $\mathbf{K}_j$ and $\mathbf{V}_j$ where the corresponding mask entry $\mathbf{M}_{ij} = 1$.

If $\mathbf{M}_{ij} = 0$, the calculation of the score matrix $\mathbf{S}_{ij}$ and the subsequent state update steps are entirely bypassed. Consequently, the state remains unchanged from the previous iteration; that is, $(\mathbf{O}_i^{(j)}, \mathbf{m}_i^{(j)}, \mathbf{l}_i^{(j)}) = (\mathbf{O}_i^{(j-1)}, \mathbf{m}_i^{(j-1)}, \mathbf{l}_i^{(j-1)})$.

# B. Detailed Analysis of Permutation Effects

In Section 2, we summarized the impact of permutation on structural densification. Here, we provide a granular breakdown of these effects across tasks, model layers, and attention heads, and offer a visual analysis of the failure modes. Unless otherwise specified, analyses are conducted using Llama-3.1-8B-Instruct with a context length of 16K tokens.

## B.1. Task-wise Proxy Sensitivity

Table 4 evaluates five permutation proxies across diverse domains, including story QA, multi-hop QA, summarization, few-shot classification, code completion, and synthetic retrieval. Random permutation consistently hurts block-level sparsity,

confirming that arbitrary reordering disrupts useful locality in the original sequence. Greedy local alignment improves over random permutation but remains weaker than attention-based global proxies. Most importantly, Random-Query-Attn, Avg-Query-Attn, and Last-Block-Query-Attn yield similar sparsity gains across tasks, including retrieval-style tasks such as NIAH and variable tracing. This supports the claim that the gain mainly comes from clustering task-agnostic heavy hitters, rather than from a brittle dependence on the specific last-query-block proxy.

*Table 4.* Task-wise sensitivity of permutation proxies. We report absolute sparsity gain $\Delta_s$ at 97.5% attention coverage with oracle block selection. Positive values indicate that permutation requires lower block density than the unpermuted baseline.

| Proxy | NarrativeQA | HotpotQA | GovReport | TREC | LCC | NIAH-Single | NIAH-Multikey | VT |
|---|---|---|---|---|---|---|---|---|
| Random | -6.11 | -9.07 | -6.67 | -7.38 | -7.45 | -6.87 | -5.31 | -7.53 |
| Greedy | 3.00 | 0.74 | 1.19 | 1.62 | 1.70 | 0.78 | 1.32 | 0.61 |
| Random-Query-Attn | **7.90** | 1.30 | 3.19 | **5.39** | 6.97 | 3.25 | 4.69 | 4.50 |
| Avg-Query-Attn | 7.73 | **2.35** | **4.22** | 4.39 | **8.02** | **3.35** | **4.79** | 4.48 |
| Last-Block-Query-Attn | 7.77 | 0.41 | 2.82 | 5.25 | 7.42 | 3.34 | 4.72 | **4.69** |

## B.2. Selected Block Counts

Figure 5 in the main text reports block-level density as a relative sparsity metric. To make this density result more concrete, Table 5 converts the density values into selected-block counts. For a sequence with $T = \lceil N/B \rceil$ blocks, the total number of causal blocks is $T(T + 1)/2$, and the selected-block count is obtained by multiplying this total by the measured block-level density. Permutation consistently reduces the block count, and the relative reduction grows from 10.7% at 8K to 14.4% at 128K. This confirms that structural consolidation becomes increasingly useful for longer contexts, where important tokens are more fragmented across the sequence.

*Table 5.* Density-equivalent selected-block counts with and without permutation. Counts are converted from the block-level density results in Figure 5 for Llama-3.1-8B with block size $B = 128$, averaged across all layers and heads at a fixed selection threshold of 0.9.

| Context | Total Causal Blocks | Selected w/o Perm | Selected w/ Perm | Reduction | Rel. |
|---|---|---|---|---|---|
| 8K | 2,080 | 1,350 | 1,205 | 145 | 10.7% |
| 16K | 8,256 | 4,457 | 4,005 | 452 | 10.1% |
| 32K | 32,896 | 15,135 | 13,337 | 1,798 | 11.9% |
| 64K | 131,328 | 43,064 | 37,632 | 5,432 | 12.6% |
| 128K | 524,800 | 134,270 | 114,996 | 19,274 | 14.4% |

## B.3. Layer-wise Sparsity Improvement

To quantify the benefit of permutation, we define **Absolute Sparsity Improvement** ($\Delta_s$) at a fixed coverage level $C$:

$$\Delta_s(C) = \text{Density}_{\text{baseline}}(C) - \text{Density}_{\text{permuted}}(C) \tag{14}$$

where $\text{Density}(C)$ is the fraction of blocks required to achieve attention mass coverage $C$. A positive $\Delta_s$ indicates that the permuted method requires fewer blocks to capture the same amount of information.

Figure 7 illustrates $\Delta_s$ across all layers at three coverage levels $(0.925, 0.950, 0.975)$. For the permutation method comparisons, the results align with the findings in Figure 1 across all layers. Strategies leveraging query attention consistently improve sparsity compared to Random Permutation and the baseline; moreover, this improvement becomes more significant at higher coverage levels. This confirms that the proposed permutation strategies effectively group critical key tokens, which is particularly beneficial given the long-tailed nature of the coverage-density distribution (Figure 1). In the layer-wise breakdown, **Layer 0 consistently exhibits high sparsity improvement**. This indicates that the "heavy hitter" phenomenon (or the "Vertical Lines" in the attention map) is especially prominent in the first layer, where permutation successfully consolidates these globally attended keys. **For the remaining layers, the sparsity improvement scales with the coverage level.** This suggests that for most of the layers (especially the middle-to-deep layers), the primary benefit of permutation stems from clustering the scattered "heavy hitter" tokens located in the tail of the attention mass distribution, which are otherwise expensive to retrieve.

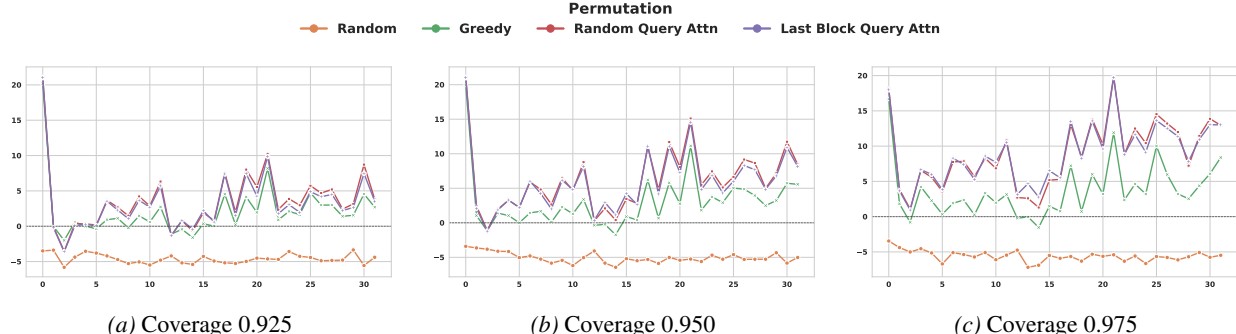

*(a)* Coverage 0.925     *(b)* Coverage 0.950     *(c)* Coverage 0.975

*Figure 7.* Layer-wise absolute sparsity improvement at various coverage levels. This metric calculates the sparsity improved by permutation. For example, if the baseline requires 60% block density to achieve 0.95 coverage, while the permuted method requires only 40%, the recorded sparsity improvement is 20%. Results are measured with Llama-3.1-8B with a context length of 16K.

*Table 6.* Head-level characterization of sparsity gain from permutation. We compute $\Delta_s$ for all 1024 heads of Llama-3.1-8B at 32K context and 97.5% attention coverage on `niah_single`.

| Category | Criterion | Count | Fraction |
|---|---|---|---|
| Helped | $\Delta_s > 1$ | 725 | 70.8% |
| Unchanged | $|\Delta_s| \leq 1$ | 246 | 24.0% |
| Harmed | $\Delta_s < -1$ | 53 | 5.2% |

## B.4. Head-wise Sparsity Improvement

To quantify when permutation helps or hurts, Table 6 aggregates the head-level sparsity gain over all 1024 heads of Llama-3.1-8B. More than 70% of heads benefit from permutation, while only 5.2% are negatively affected. The harmed heads concentrate in early layers (layers 3–6, 16.1% harmed), whereas middle-to-deep layers (layers 7–18) show near-universal improvement, with 89.6% helped and only 0.8% harmed. This confirms that the aggregate gain is not driven by a few outlier heads; instead, permutation is broadly beneficial, with failures localized to specific attention patterns analyzed in Section B.5.

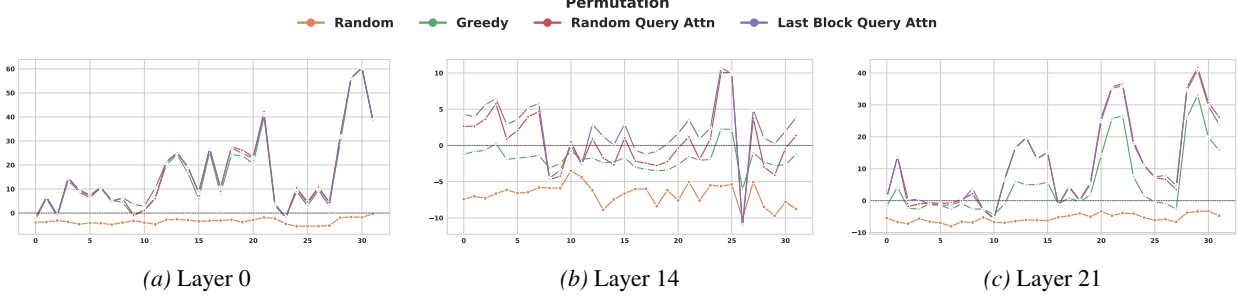

*(a)* Layer 0     *(b)* Layer 14     *(c)* Layer 21

*Figure 8.* Head-wise absolute sparsity improvement of representative layers at attention coverage of 0.975.

Figure 8 shows the absolute head-wise sparsity improvement for three representative layers with an attention coverage of 0.975. The results reveal diverse responses to permutation across layers and heads. In the first layer, which consistently benefits from permutation, nearly all heads become sparser (Figure 8a), with some showing substantial gains (e.g., Head 30 shows a 60% absolute improvement). In most other layers, the vast majority of heads improve with permutation. However, we identified a few outliers; for example, Head 26 in Layer 14 (Figure 8b) becomes denser, resulting in only a marginal overall improvement for that layer. In contrast, other layers like Layer 21 (Figure 8c) lack these negatively affected heads, and their mix of insensitive and improved heads leads to a noticeable overall increase in sparsity.

## B.5. Failure Mode Analysis

Here we analyze why certain attention heads exhibit marginal improvement or even degraded sparsity under permutation by visualizing their attention maps. Zooming in on Figure 8, we select two representative cases: Layer 14 Head 26 (negative

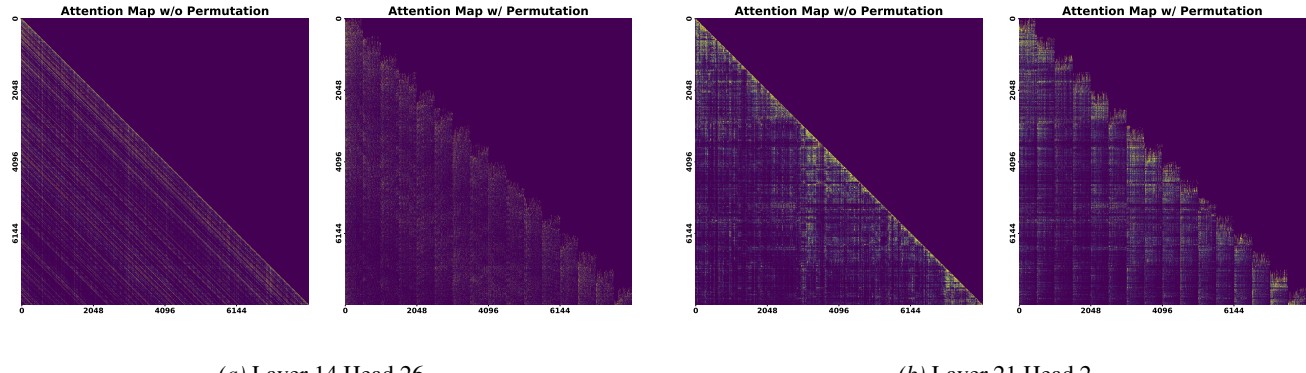

*(a)* Layer 14 Head 26                    *(b)* Layer 21 Head 2

*Figure 9.* Visualization of attention maps for heads without noticeable sparsity gains.

sparsity gain) and Layer 21 Head 2 (marginal gain). For the minority of heads dominated by the "Slash Line" pattern (Figure 9a), a phenomenon also recognized in previous literature (Jiang et al., 2024; Lai et al., 2025) where queries attend to keys at fixed intervals, permutation fails to improve sparsity. This occurs because selecting the corresponding diagonal blocks is naturally the optimal strategy to cover "Slash Lines". Any permutation inevitably disrupts this structure, scattering the keys and leading to redundancy in block selection. Regarding heads showing highly query-specific patterns where different queries attend to distinct sets of keys (Figure 9b), the sparsity improvement from permutation remains marginal. In contrast, permutation yields significant improvements for the majority of heads where most queries attend to the same shared set of keys (Figure 3). Consequently, the overall sparsity improvement could be further enhanced by incorporating pruning strategies to exclude the few heads with negative sparsity gains, which we leave for future work.

## C. Proofs of Permutation Properties

### C.1. Proof of Lemma 3.1

**Lemma C.1** (Key-Value Pair Permutation Invariance). *The attention mechanism is invariant to the order of the source sequence, provided that the key-value pairings are maintained.*

*Formally, let $\mathbf{P}_\pi \in \{0,1\}^{M \times M}$ be a permutation matrix that reorders the rows of a matrix according to a permutation $\pi$ on the index set $\{1, \ldots, M\}$. The following identity holds:*

$$Attention(\mathbf{Q}, \mathbf{P}_\pi \mathbf{K}, \mathbf{P}_\pi \mathbf{V}) = Attention(\mathbf{Q}, \mathbf{K}, \mathbf{V}) \tag{15}$$

*Proof.* Let $\mathbf{O} = \text{Attention}(\mathbf{Q}, \mathbf{K}, \mathbf{V})$ and $\mathbf{O}' = \text{Attention}(\mathbf{Q}, \mathbf{P}_\pi \mathbf{K}, \mathbf{P}_\pi \mathbf{V})$. Our goal is to show that $\mathbf{O} = \mathbf{O}'$. We will prove this by showing that their corresponding row vectors, $\mathbf{o}_i$ and $\mathbf{o}'_i$, are equal for any arbitrary row index $i \in \{1, \ldots, N\}$.

Let $\mathbf{A} = \frac{\mathbf{Q}\mathbf{K}^T}{\sqrt{d}}$ and $\mathbf{W} = \text{softmax}(\mathbf{A})$(we use W instead of P as in Eq.9 to avoid confusion) The $i$-th row of the original output is given by:

$$\mathbf{o}_i = \sum_{j=1}^{M} \mathbf{W}_{ij} \mathbf{v}_j$$

Now, let $\mathbf{K}' = \mathbf{P}_\pi \mathbf{K}$ and $\mathbf{V}' = \mathbf{P}_\pi \mathbf{V}$. The score matrix for $\mathbf{O}'$ is $\mathbf{A}' = \frac{\mathbf{Q}(\mathbf{K}')^T}{\sqrt{d}} = \frac{\mathbf{Q}\mathbf{K}^T \mathbf{P}_\pi^T}{\sqrt{d}} = \mathbf{A}\mathbf{P}_\pi^T$. Let $\mathbf{W}' = \text{softmax}(\mathbf{A}')$.

The $(i, j)$-th element of $\mathbf{A}'$ is $\mathbf{A}'_{ij} = \sum_{l=1}^{M} \mathbf{A}_{il}(\mathbf{P}_\pi^T)_{lj} = \mathbf{A}_{i, \pi^{-1}(j)}$. The denominator for the softmax computation on the $i$-th row of $\mathbf{A}'$ is:

$$\sum_{l=1}^{M} \exp(\mathbf{A}'_{il}) = \sum_{l=1}^{M} \exp(\mathbf{A}_{i, \pi^{-1}(l)})$$

Since $\pi^{-1}$ is a bijection on $\{1, \ldots, M\}$, this summation is a reordering of the terms $\sum_{k=1}^{M} \exp(A_{ik})$, which is the denominator for the $i$-th row of the original weights $W$.

Thus, the $(i, j)$-th element of the new weight matrix $W'$ is:

$$\mathbf{W}'_{ij} = \frac{\exp(\mathbf{A}'_{ij})}{\sum_{l=1}^{M} \exp(\mathbf{A}'_{il})} = \frac{\exp(\mathbf{A}_{i,\pi^{-1}(j)})}{\sum_{k=1}^{M} \exp(\mathbf{A}_{ik})} = \mathbf{W}_{i,\pi^{-1}(j)}$$

The $i$-th row of the new output $O'$ is a weighted sum of the rows of $V' = P_\pi V$. The $j$-th row of $V'$ is $v'_j = v_{\pi^{-1}(j)}$. Therefore:

$$\mathbf{o}'_i = \sum_{j=1}^{M} \mathbf{W}'_{ij} \mathbf{v}'_j = \sum_{j=1}^{M} \mathbf{W}_{i,\pi^{-1}(j)} \mathbf{v}_{\pi^{-1}(j)}$$

Let $k = \pi^{-1}(j)$. Since $\pi^{-1}$ is a bijection, summing over all $j \in \{1, \ldots, M\}$ is equivalent to summing over all $k \in \{1, \ldots, M\}$. By this change of variables, we have:

$$\mathbf{o}'_i = \sum_{k=1}^{M} \mathbf{W}_{ik} \mathbf{v}_k = \mathbf{o}_i$$

Since $\mathbf{o}'_i = \mathbf{o}_i$ for an arbitrary $i$, the matrices $\mathbf{O}'$ and $\mathbf{O}$ are identical. □

### C.2. Proof of Lemma 3.2

**Lemma C.2** (Query Permutation Equivariance). *The attention mechanism is equivariant with respect to permutations of the query sequence.*

*Formally, let $\mathbf{P}_\sigma \in \{0, 1\}^{N \times N}$ be a permutation matrix that reorders the rows of a matrix according to a permutation $\sigma$ on the index set $\{1, \ldots, N\}$. The following relationship holds:*

$$Attention(\mathbf{P}_\sigma \mathbf{Q}, \mathbf{K}, \mathbf{V}) = \mathbf{P}_\sigma Attention(\mathbf{Q}, \mathbf{K}, \mathbf{V}) \tag{16}$$

*Proof.* Let $\mathbf{O} = \text{Attention}(\mathbf{Q}, \mathbf{K}, \mathbf{V})$ and $\mathbf{O}' = \text{Attention}(\mathbf{P}_\sigma \mathbf{Q}, \mathbf{K}, \mathbf{V})$. We want to show that $\mathbf{O}' = \mathbf{P}_\sigma \mathbf{O}$.

Let $\mathbf{A} = \frac{\mathbf{Q}\mathbf{K}^T}{\sqrt{d}}$ and $\mathbf{W} = \text{softmax}(\mathbf{A})$, such that $\mathbf{O} = \mathbf{W}\mathbf{V}$. The score matrix for $\mathbf{O}'$ is $\mathbf{A}' = \frac{(\mathbf{P}_\sigma \mathbf{Q})\mathbf{K}^T}{\sqrt{d}} = \mathbf{P}_\sigma \left( \frac{\mathbf{Q}\mathbf{K}^T}{\sqrt{d}} \right) = \mathbf{P}_\sigma \mathbf{A}$. Let $\mathbf{W}' = \text{softmax}(\mathbf{A}')$.

The softmax function operates independently on each row. Let $(\mathbf{X})_i$ denote the $i$-th row of a matrix $\mathbf{X}$. Left-multiplication by $\mathbf{P}_\sigma$ permutes the rows of $\mathbf{A}$, such that the $i$-th row of $\mathbf{A}'$ is the $\sigma^{-1}(i)$-th row of $\mathbf{A}$: $(\mathbf{A}')_i = (\mathbf{A})_{\sigma^{-1}(i)}$. Applying the softmax function, the $i$-th row of $\mathbf{W}'$ is:

$$(\mathbf{W}')_i = \text{softmax}((\mathbf{A}')_i) = \text{softmax}((\mathbf{A})_{\sigma^{-1}(i)})$$

This resulting vector is identical to the $\sigma^{-1}(i)$-th row of the original weight matrix $\mathbf{W}$. Thus, $(\mathbf{W}')_i = (\mathbf{W})_{\sigma^{-1}(i)}$. This equality for all rows $i$ implies that the entire matrix $\mathbf{W}'$ is a row-permuted version of $\mathbf{W}$, i.e., $\mathbf{W}' = \mathbf{P}_\sigma \mathbf{W}$.

Now we can write the output $\mathbf{O}'$ as:

$$\mathbf{O}' = \mathbf{W}'\mathbf{V} = (\mathbf{P}_\sigma \mathbf{W})\mathbf{V}$$

By the associativity of matrix multiplication, we have:

$$\mathbf{O}' = \mathbf{P}_\sigma(\mathbf{W}\mathbf{V}) = \mathbf{P}_\sigma \mathbf{O}$$

This completes the proof. □

### C.3. Proof of Theorem 3.3

**Theorem C.3** (Attention Permutation Invariance under Inverse Transformation). *If the queries are permuted by $\mathbf{P}_\sigma$ and the key-value pairs are permuted by $\mathbf{P}_\pi$, the resulting output is a permuted version of the original output. Applying the inverse of the query permutation recovers the original, unpermuted output. Formally:*

$$\mathbf{P}_\sigma^T Attention(\mathbf{P}_\sigma \mathbf{Q}, \mathbf{P}_\pi \mathbf{K}, \mathbf{P}_\pi \mathbf{V}) = Attention(\mathbf{Q}, \mathbf{K}, \mathbf{V}) \tag{17}$$

*Proof.* We prove the theorem by showing that the left-hand side (LHS) of the equation simplifies to the right-hand side (RHS) through sequential application of the preceding lemmas.

$$
\begin{aligned}
\text{LHS} &= \mathbf{P}_\sigma^T \, \text{Attention}(P_\sigma Q, P_\pi K, P_\pi V) \\
&= \mathbf{P}_\sigma^T \, \text{Attention}(\mathbf{P}_\sigma Q, K, V) && \text{by Lemma 3.1} \\
&= \mathbf{P}_\sigma^T \, (\mathbf{P}_\sigma \, \text{Attention}(Q, K, V)) && \text{by Lemma 3.2} \\
&= (\mathbf{P}_\sigma^T \mathbf{P}_\sigma) \, \text{Attention}(Q, K, V) && \text{by associativity} \\
&= I \cdot \text{Attention}(Q, K, V) && \text{since } P_\sigma \text{ is orthogonal} \\
&= \text{Attention}(Q, K, V) \\
&= \text{RHS}
\end{aligned}
$$

The final expression is identical to the right-hand side, which concludes the proof. □

# D. Block Selection

## D.1. Block Selection in PBS-Attn

We use a mean pooling strategy and block-wise attention to estimate the importance of each key block. This method is also used for unpermuted sequences, serving as a strong baseline denoted as MeanPooling in the main paper. Here we detail the implementation of MeanPooling selection in Algorithm 2. Note that for the baseline MeanPooling, $\mathbf{Q}'$ and $\mathbf{K}'$ remain unpermuted as $\mathbf{Q}' = \mathbf{Q}$ and $\mathbf{K}' = \mathbf{K}$. The causal mask $\mathbf{C}$ is a upper triangular matrix with entries set to $-\infty$. If segmented permutation is applied, this mask also includes the on-diagonal segments (as in Figure 2), to ensure valid intra-segment attention post-permutation.

---

**Algorithm 2** MeanPooling Block Selection

---

**Require:** Query matrix $\mathbf{Q}' \in \mathbb{R}^{N \times d}$, Key matrix $\mathbf{K}' \in \mathbb{R}^{N \times d}$, block size $B$, attention score threshold $\tau$, causal mask $\mathbf{C} \in \{0, -\infty\}^{\lceil N/B \rceil \times \lceil N/B \rceil}$.
**Ensure:** Block selection mask $\mathbf{M} \in \{0, 1\}^{\lceil N/B \rceil \times \lceil N/B \rceil}$.
 1: Divide $\mathbf{Q}', \mathbf{K}'$ into blocks of size $B$: $\{\mathbf{Q}'_i\}_{i=1}^{T_r}, \{\mathbf{K}'_j\}_{j=1}^{T_c}$, where $T_r = T_c = \lceil N/B \rceil$.
 2: Compute pooled queries: $\bar{\mathbf{Q}}_i = \text{MeanPool}(\mathbf{Q}'_i)$ for $i = 1, \ldots, T_r$.
 3: Compute pooled keys: $\bar{\mathbf{K}}_j = \text{MeanPool}(\mathbf{K}'_j)$ for $j = 1, \ldots, T_c$.
 4: Form pooled matrices $\bar{\mathbf{Q}} \in \mathbb{R}^{T_r \times d}$ and $\bar{\mathbf{K}} \in \mathbb{R}^{T_c \times d}$.
 5: Compute block scores: $\mathbf{S}_{\text{block}} = \text{softmax}(\bar{\mathbf{Q}}\bar{\mathbf{K}}^T/\sqrt{d} + \mathbf{C})$.
 6: Initialize $\mathbf{M} = \mathbf{0}$.
 7: **for** $i = 1$ to $T_r$ **do**
 8:     Get scores for query block $i$: $\mathbf{a}_i = \mathbf{S}_{\text{block}}[i, 1:i]$.
 9:     Sort scores and get original indices: $\mathbf{o}_i = \text{argsort}(-\mathbf{a}_i)$.
10:     Compute cumulative sum on sorted scores: $\mathbf{c}_i = \text{cumsum}(\mathbf{a}_i[\mathbf{o}_i])$.
11:     Find number of blocks to select: $k = \min(\{j \mid \mathbf{c}_i[j] \geq \tau\} \cup \{i\})$.
12:     Get indices of blocks to select: $\mathcal{J} = \mathbf{o}_i[1:k]$.
13:     Set $\mathbf{M}[i, j] = 1$ for all $j \in \mathcal{J}$.
14: **end for**
15: **return** $\mathbf{M}$.

---

## D.2. PBS-Attn with Existing Block Selection Algorithms

In the main paper, we use a simple mean pooling strategy for block selection in block-sparse attention, as detailed in Section D.1, and show that permutation can increase block-level sparsity under this naive mean pooling strategy (Section 4.3). In this section, we further demonstrate that advanced block selection algorithms (e.g. XAttention) can also benefit from permutation.

As shown in Figure 10, XAttention selection can also benefit from the sparsity improvements of permutation, achieving a better trade-off between performance and sparsity.

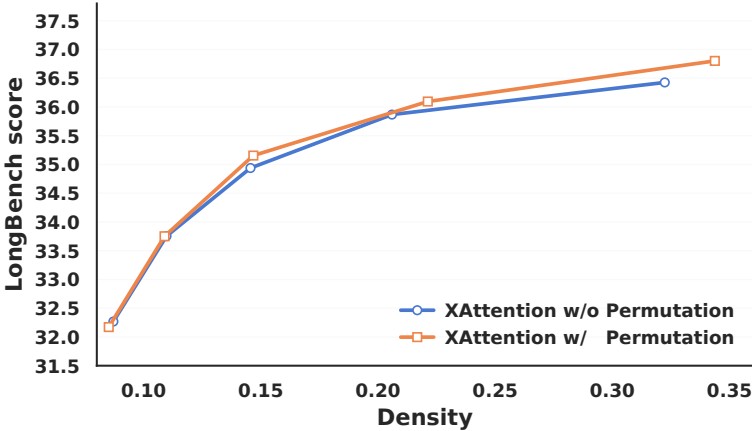

*Figure 10.* Longbench score vs. average block-level density at a context length of 32k of XAttention selection with and without permutation.

## E. Analysis on the Permutation Overhead

**Time Overhead**    As shown in Figures 11a and 11b, the permutation overhead in PBS-Attn is negligible compared to the main attention computation time, especially at longer context lengths. For instance, at a context length of 128K, permutation introduces an overhead of only $4\%$ relative to the block attention computation time and just $1.3\%$ compared to FlashAttention. While permuting queries introduces a slightly higher overhead than permuting keys, this difference diminishes as the context length increases. However, query permutation can also result in lower block-level sparsity than key permutation under the same settings, leading to higher attention computation time. Detailed benchmarking results are shown in Table 7. At a 512K context length, the permutation overhead (i.e., permutation time plus block selection time) is only $3.1\%$, while providing a $3.41\times$ attention speedup, further demonstrating the practical potential of PBS-Attn.

*Table 7.* Timing breakdown (ms) for PBS-Attn relative to FlashAttention. Measured by profiling on CUDA events on an H100 80GB GPU.

| Length | FlashAttention | Permutation | Block Selection | Attention | Total | Overhead | Speedup |
|---|---|---|---|---|---|---|---|
| | | | *Key Permutation* | | | | |
| **4K** | 0.54 | 0.72 | 0.84 | 0.53 | 2.09 | 74.6% | 0.26× |
| **8K** | 1.78 | 0.82 | 0.86 | 1.31 | 2.99 | 56.2% | 0.60× |
| **16K** | 6.67 | 1.02 | 0.62 | 4.08 | 5.71 | 28.7% | 1.17× |
| **32K** | 26.10 | 1.68 | 0.60 | 13.83 | 16.11 | 14.2% | 1.62× |
| **64K** | 106.84 | 3.24 | 1.24 | 42.44 | 46.92 | 9.5% | 2.28× |
| **128K** | 443.29 | 6.22 | 3.21 | 150.45 | 159.87 | 5.9% | 2.77× |
| **256K** | 1837.32 | 13.87 | 11.81 | 563.54 | 589.22 | 4.4% | 3.12× |
| **512K** | 7496.67 | 26.76 | 41.99 | 2128.16 | 2196.91 | 3.1% | 3.41× |
| | | | *Query Permutation* | | | | |
| **4K** | 0.54 | 0.70 | 0.81 | 0.63 | 2.31 | 65.4% | 0.23× |
| **8K** | 1.78 | 0.82 | 0.81 | 1.67 | 3.30 | 49.4% | 0.54× |
| **16K** | 6.67 | 1.20 | 0.49 | 5.06 | 6.75 | 25.0% | 0.99× |
| **32K** | 26.10 | 1.98 | 0.59 | 17.81 | 20.38 | 12.6% | 1.28× |
| **64K** | 106.84 | 3.52 | 1.25 | 52.95 | 57.72 | 8.3% | 1.85× |
| **128K** | 443.29 | 7.10 | 3.23 | 181.51 | 191.85 | 5.4% | 2.31× |
| **256K** | 1837.32 | 13.82 | 11.80 | 597.04 | 622.66 | 4.1% | 2.95× |
| **512K** | 7496.67 | 26.83 | 41.91 | 2481.68 | 2550.42 | 2.7% | 2.94× |

**Memory Overhead**    Here we analyze the memory overhead of PBS-Attn. For the proposed Last-Block-Query Key Permutation strategy, we require only the last block of queries to calculate proxy scores; consequently, the memory cost for scoring scales linearly with context length. Specifically, given block size $B$ and context length $N$, the memory cost for

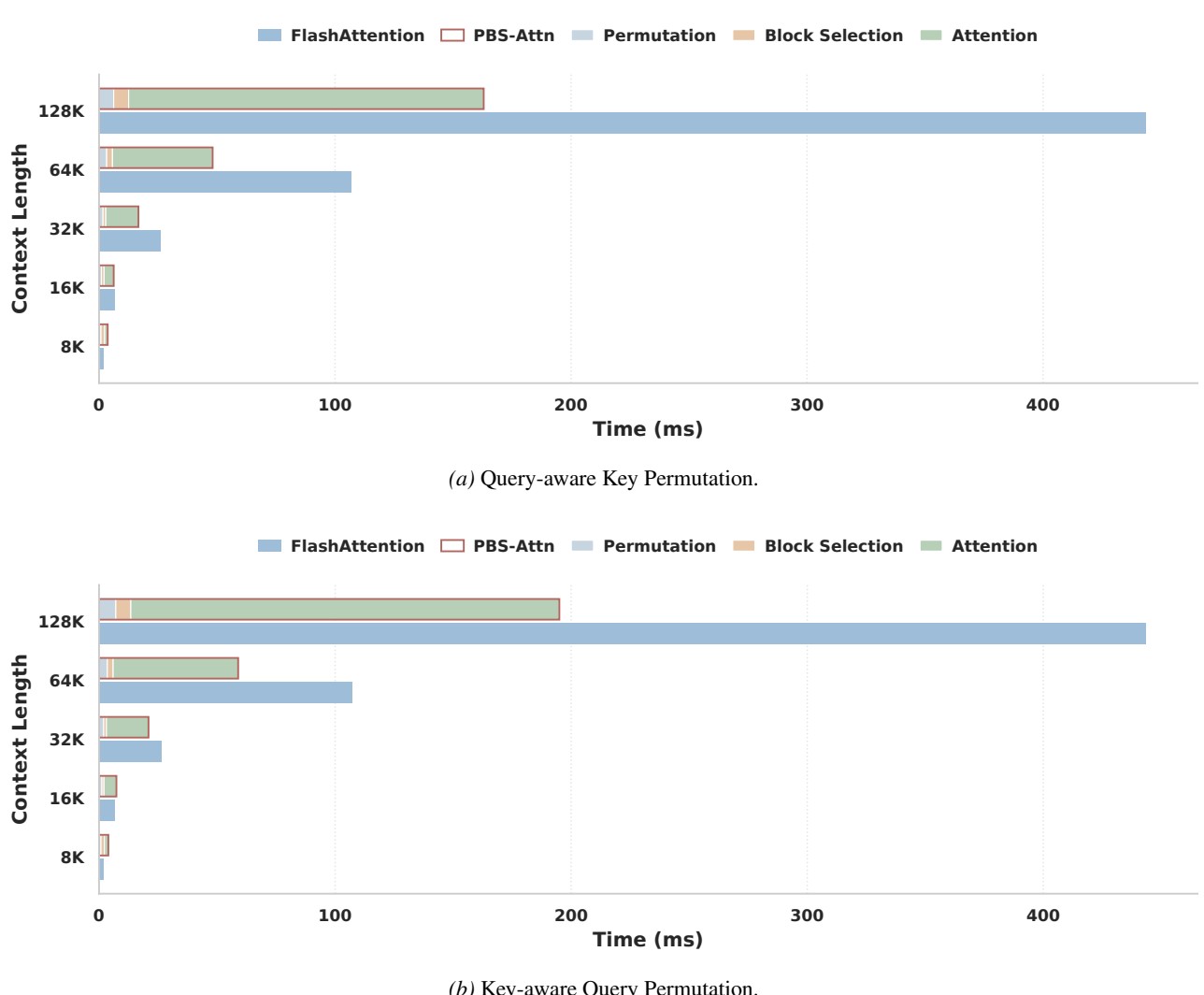

*(a)* Query-aware Key Permutation.

*(b)* Key-aware Query Permutation.

*Figure 11.* Detailed benchmarking results of PBS-Attn vs. FlashAttention.

scoring is $O(B \times N)$. For $B = 128$ and $N = 128$K, this amounts to approximately 32MiB per head in BFloat16, which is negligible relative to the total activation memory for long sequences. Regarding the memory overhead during permutation, our current implementation explicitly creates physically permuted key and value tensors using a `torch.gather()` call, which allocates a temporary buffer proportional to the sequence size. For Llama-3.1-8B with head dimension $d = 128$, this results in an additional 32MiB per head in BFloat16, which is also insignificant. Furthermore, this overhead could be mitigated via index remapping, allowing the attention kernel to retrieve data directly from the original vectors using permuted indices. Nonetheless, since the prefilling phase of LLMs is primarily compute-bound, the impact of this memory movement is minimal. As shown in Table 7, the relative overhead decreases further in memory-intensive longer context scenarios. Figure 12 further demonstrates that PBS-Attn maintains consistent speedups on larger models (e.g., Qwen-2.5-14B) despite their higher memory requirements, confirming that memory overhead does not become a bottleneck as model size scales.

## F. Additional Model Evaluations

### F.1. Evaluation on Qwen3-8B

To further validate the generalizability of PBS-Attn on newer model families, we evaluate Qwen3-8B (Qwen Team, 2025) on LongBench. All hyperparameters are kept identical to the main experiments. As shown in Table 8, PBS-Attn matches the full-attention baseline in average performance (33.98 vs. 34.08) and remains competitive with strong sparse-attention baselines. We further measure end-to-end TTFT across context lengths in Table 9. The speedup trajectory is consistent with

the main Llama-3.1-8B results: PBS-Attn becomes increasingly beneficial as the context length grows, reaching a $2.72\times$ speedup at 256K context. At 512K, tensor-parallel communication overhead reduces the relative gain, but PBS-Attn still delivers a practical $1.92\times$ end-to-end speedup.

*Table 8.* Performance comparison of various sparse attention methods on LongBench with Qwen3-8B. **Bold** and underlined scores indicate the best and second-best performing sparse methods in each category, respectively, with the exception of the full attention baseline.

| Method | Single-Doc QA | Multi-Doc QA | Summarization | Few-shot Learning | Code | Synthetic | Avg. |
|---|---|---|---|---|---|---|---|
| Full | 46.93 | 37.97 | 16.57 | 34.07 | 2.93 | 66.00 | 34.08 |
| MInference | 46.92 | 37.11 | 16.53 | 34.15 | **2.98** | 66.17 | **33.98** |
| FlexPrefill | 46.08 | 37.29 | 16.53 | 33.63 | 2.52 | 49.00 | 30.84 |
| XAttention | 45.66 | **37.77** | **16.66** | 36.20 | 2.02 | 64.67 | 33.83 |
| MeanPooling | 45.69 | 37.25 | 16.61 | **37.45** | 2.52 | 62.33 | 33.64 |
| **PBS-Attn** | **47.04** | 37.17 | **16.66** | 33.88 | 2.80 | **66.33** | **33.98** |

*Table 9.* End-to-end TTFT comparison on Qwen3-8B across context lengths. Latency is reported in milliseconds. Tensor parallelism is used for the 256K and 512K contexts to fit memory.

| Method | 8K | 16K | 32K | 64K | 128K | 256K (tp=4) | 512K (tp=8) |
|---|---|---|---|---|---|---|---|
| Full | 326.8 | 732.7 | 1905.3 | 5573.0 | 18628.0 | 20104.7 | 53999.5 |
| **PBS-Attn** | 368.9 | 715.0 | 1536.4 | 3540.4 | 8396.0 | 7400.9 | 28136.1 |
| Speedup | 0.89× | 1.02× | 1.24× | 1.57× | 2.22× | 2.72× | 1.92× |

## F.2. Evaluation on Qwen-2.5-14B-1M

To further validate the effectiveness of PBS-Attn on larger LLMs, we conduct evaluations using Qwen-2.5-14B-1M on the LongBench benchmark. As presented in Table 10, PBS-Attn consistently outperforms the baselines; this is consistent with the results observed in Table 1 and confirms the scalability of PBS-Attn to larger LLMs. Regarding efficiency, Figure 12 illustrates that PBS-Attn achieves nearly a $2\times$ speedup at a context length of 128K. This trajectory closely matches that of the 7B model, further demonstrating the method's efficiency at scale. Note that the speedup at the 256K context length cannot be directly compared to the 7B model results due to differing tensor parallelism settings required by memory constraints.

*Table 10.* Performance comparison of various sparse attention methods on LongBench with Qwen-2.5-14B-1M. **Bold** and underlined scores indicate the best and second-best performing methods in each category, respectively, with the exception of the full attention baseline.

| Method | Single-Doc QA | Multi-Doc QA | Summarization | Few-shot Learning | Code | Synthetic | Avg. |
|---|---|---|---|---|---|---|---|
| Full | 47.33 | 47.44 | 15.55 | 59.13 | 18.67 | 71.33 | 43.24 |
| MInference | 46.50 | 45.73 | 15.56 | 57.23 | 18.76 | 63.33 | 41.19 |
| FlexPrefill | 44.73 | 43.37 | 15.62 | 53.55 | 9.88 | 35.00 | 33.69 |
| XAttention | **46.65** | 46.03 | **15.63** | **58.69** | **19.56** | 63.33 | 41.65 |
| MeanPooling | 45.41 | 45.57 | 15.58 | 57.40 | 17.25 | 35.56 | 36.13 |
| **PBS-Attn** | 46.56 | **46.43** | 15.48 | 58.51 | 17.53 | **67.17** | **41.95** |

## G. GQA Handling

Modern LLMs often employ Grouped Query Attention (GQA), where a group of query heads shares the same key and value heads to reduce inference overhead. In this section, we compare two different GQA handling strategies for PBS-Attn: (1) the **default strategy**, where we replicate the keys and values for each query head in a group to apply unique permutations, and (2) the **shared permutation strategy**, where we average the queries across the head dimension within each group to compute a single permutation, ensuring that all queries within the same group share the keys and values in the same order.

Figure 13 illustrates the trade-off between attention coverage and density for these two GQA handling strategies. The results imply that sharing the permutation within a GQA group affects sparsity only marginally, while still maintaining a significant coverage gain compared to the no-permutation baseline. We further evaluate this approach on real-world datasets using

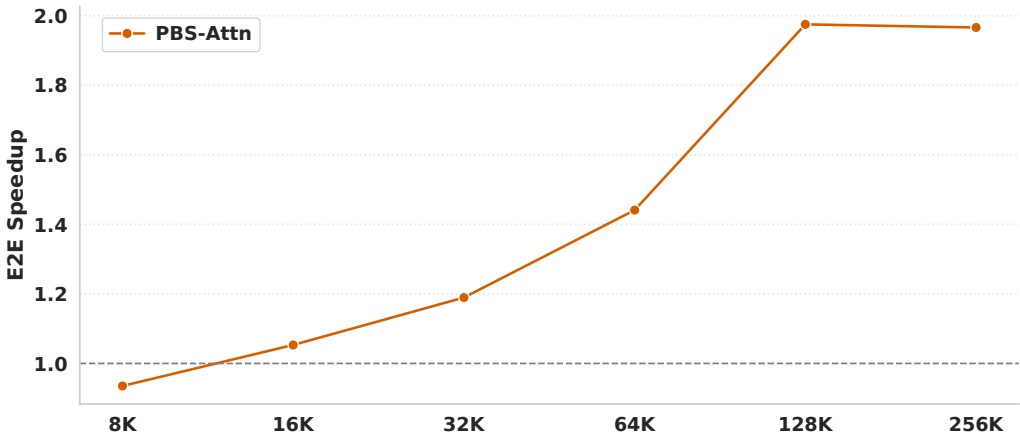

*Figure 12.* Speedup of PBS-Attn relative to FlashAttention on various context lengths for Qwen-2.5-14B-1M. We employ tensor parallelism with tp_size of 4 for 256K context due to memory constraints.

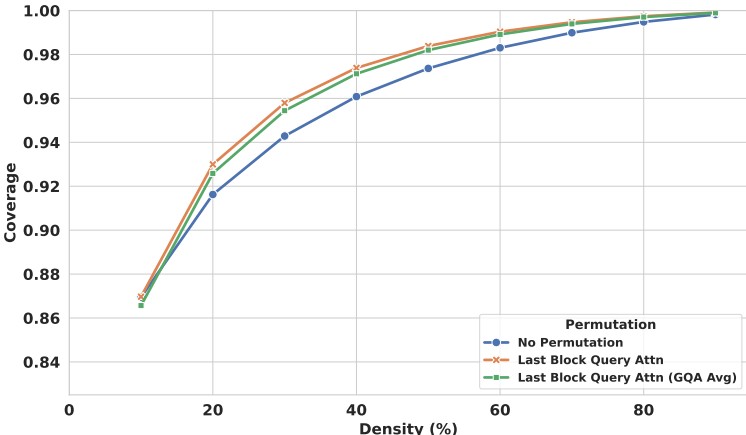

*Figure 13.* Coverage-density trade-off with two different GQA handling strategies. The results are measured with Llama-3.1-8B-Instruct with a context length of 16K.

LongBench. As shown in Table 11, the shared permutation strategy achieves performance comparable to the default strategy, closely aligning with the findings in Figure 13. This demonstrates that sharing the permutation within a GQA group has minimal impact on sparsity gains and practical performance, suggesting a more efficient approach for the deployment of PBS-Attn.

*Table 11.* Performance comparison with different GQA handling strategies on LongBench.

| Method | Single-Doc QA | Multi-Doc QA | Summarization | Few-shot Learning | Code | Synthetic | Avg. |
|---|---|---|---|---|---|---|---|
| **PBS-Attn(Default)** | 48.00 | 42.09 | 17.72 | 28.36 | 24.25 | 63.80 | 37.37 |
| **PBS-Attn(Shared Permutation)** | 48.38 | 41.47 | 17.84 | 27.77 | 23.86 | 63.92 | 37.21 |

## H. Visualization of Permutation

In this section, we provide more visualizations of the permutation effect on both Llama-3.1-8B (Figure 14) and Qwen-2.5-7B-1M (Figure 15).

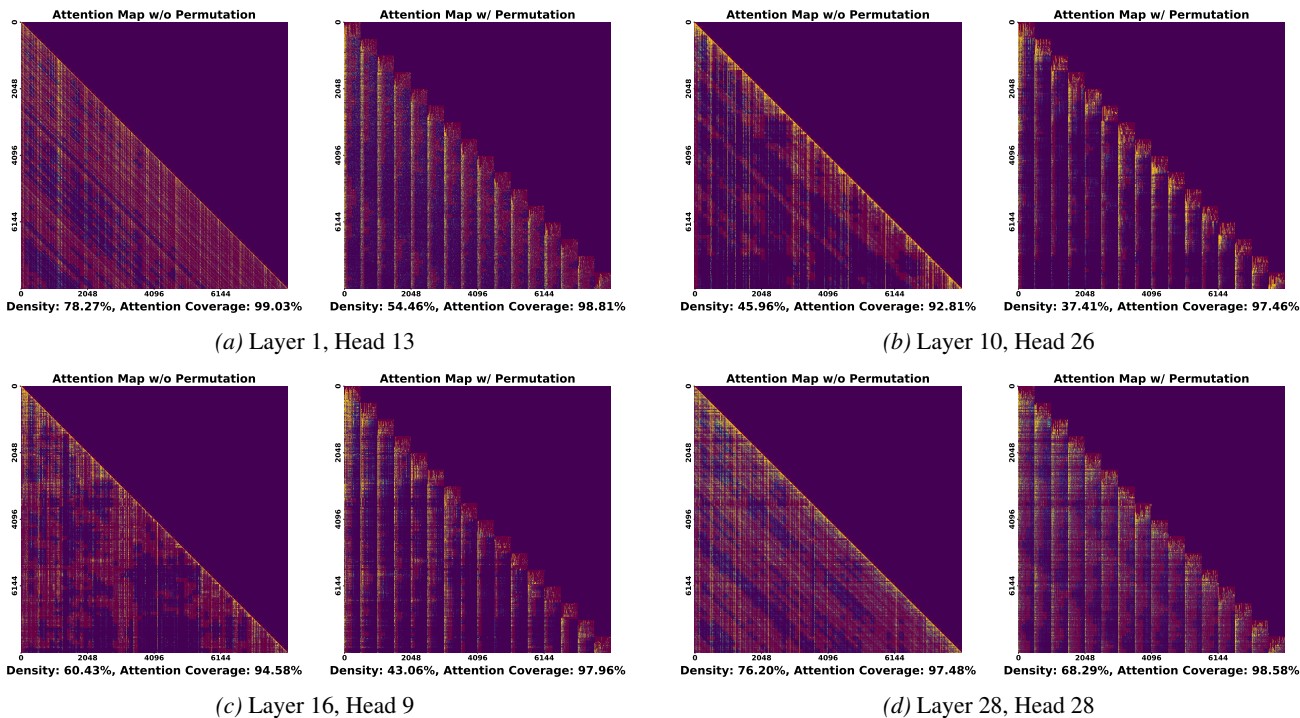

*Figure 14.* Permutation visualizations of Llama-3.1-8B.

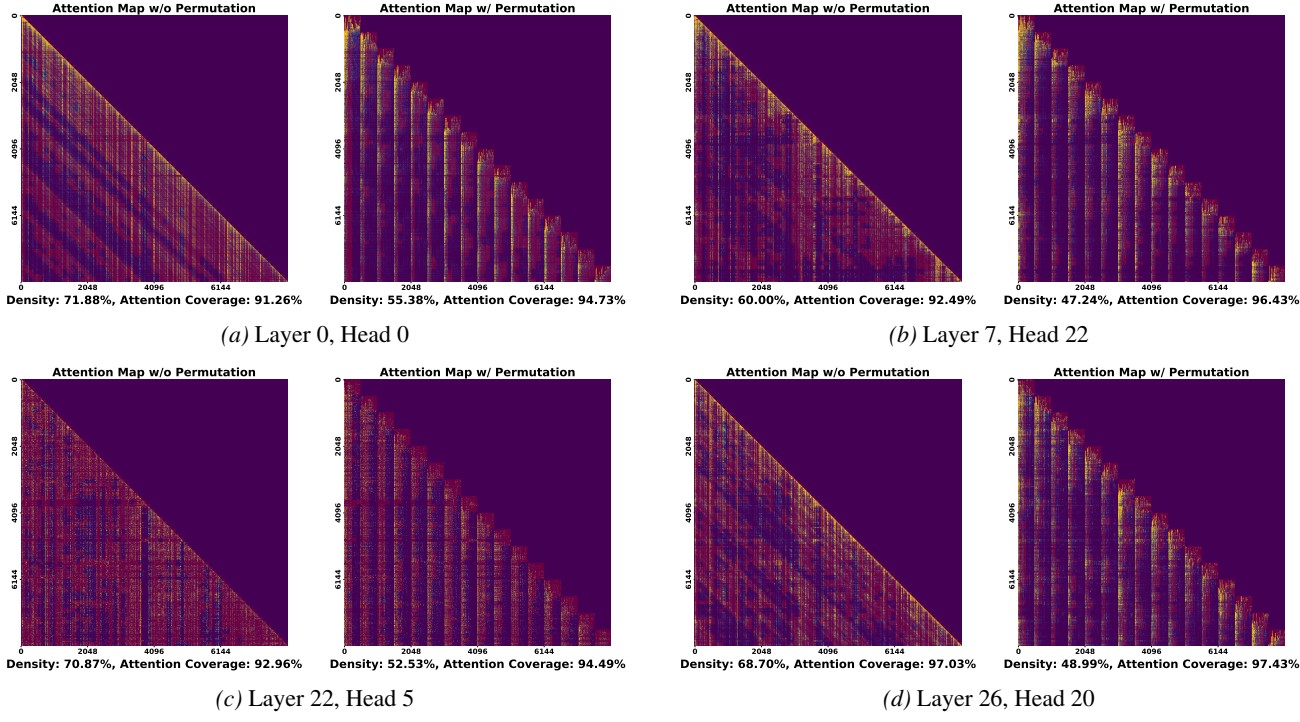

*Figure 15.* Permutation visualizations of Qwen-2.5-7B-1M.

