# OpenReview forum: "Sparser Block-Sparse Attention via Token Permutation"
_ICML.cc/2026/Conference — ICML 2026 regular_

### Official Review · Reviewer_8twW · 2026-03-10

**Soundness:** 3
**Presentation:** 3
**Significance:** 3
**Originality:** 3
**Overall Recommendation:** 4
**Confidence:** 3

**Summary:**

This paper proposes Permuted Block-Sparse Attention (PBS-Attn), a training-free method for accelerating LLM prefilling under long contexts. The key idea is to exploit permutation symmetries of attention to reorder tokens before block selection, so that globally important keys become more spatially concentrated and therefore easier to capture with fewer blocks. To preserve causality, the paper introduces segmented permutation, which only permutes tokens within segments while keeping inter-segment order fixed. The final system combines this permutation step with block-sparse attention and a custom permuted-FlashAttention kernel.

**Compliance With Llm Reviewing Policy:**

Affirmed.

**Key Questions For Authors:**

1. How stable is the last-block query proxy across tasks and domains? The method estimates global key importance using only the last query block, and Figure 1 suggests that this works about as well as using a random subset of queries. Still, this is a fairly strong assumption. I would like to see a more systematic sensitivity study across datasets or head types, especially for cases where attention is highly query-specific.

2. Can the paper better quantify when permutation helps and when it hurts? The appendix provides useful head-wise and failure-mode visualizations, but the main paper could benefit from a more explicit characterization: for example, what fraction of heads/layers are helped, unchanged, or harmed? This would make the method’s applicability much clearer.

**Limitations:**

Yes

**Strengths And Weaknesses:**

Strengths

1. Most related block-sparse methods operate on the original attention layout and focus on better block scoring or selection. This paper instead proposes to rearrange tokens first so that sparsity becomes more favorable at the block level. That framing is simple, original, and well motivated by the “information fragmentation” analysis in Section 2.

2. A naive global permutation would destroy the lower-triangular causal structure. The segmented permutation idea is a practical compromise that preserves inter-segment causality while allowing intra-segment reordering. This is an important technical detail, and the visualizations in Figures 2 and 3 help explain why the method works.

3. The analysis section is useful. The paper studies permutation target choice, segment size, block size, sparsity improvement across context lengths, layer-wise/head-wise behavior, and failure modes. I especially appreciated that the appendix does not pretend the method helps every head equally; it explicitly identifies heads with “slash-line” or highly query-specific patterns as difficult cases.

Weaknesses

1. The paper itself shows that some heads do not benefit, and in certain cases permutation can even degrade sparsity. In particular, for “slash-line” heads, the appendix argues that permutation may actually disrupt an already favorable structure. That is an important limitation, and I think it should be discussed more prominently in the main paper rather than mostly in the appendix.

2. The authors argue that permutation is agnostic to the choice of block selection algorithm, and Appendix D.2 gives evidence that XAttention also benefits. However, the main text still centers its method around a mean-pooling selector, and the evidence for broad compatibility is somewhat limited. Since a large portion of the final gains may depend on the interaction between permutation and block scoring, I would have liked a stronger and more systematic comparison across multiple selectors in the main results.

3. The RULER results reveal an important caveat.
The authors note that, for synthetic UUID-style retrieval tasks, mean-pooling-based block scoring performs poorly enough that token-level scoring is needed, leading the authors to introduce PBS-Attn+ with XAttention-style antidiagonal scoring. This suggests that the proposed permutation alone is not sufficient in some challenging settings, and its effectiveness may depend on a better scoring mechanism than the one used in the main method.

---

> ### Author Rebuttal · Authors · 2026-03-30
>
> We sincerely thank the reviewer for the detailed and nuanced evaluation. We address each concern below.
>
> **Response to W1/Q2:**
>
> We agree and will move this limitation from the Appendix to the main text. To directly quantify when permutation helps vs. hurts, we compute $\Delta_s$ per head across all 32 layers of Llama-3.1-8B (1024 heads total) at 32K context and 97.5% coverage, measured on `niah_single` task. We use `niah_single` as a stress test because this retrieval task is where the reviewers are most concerned that our permutation proxy might fail.
>
> | Category | Criterion | Count | Fraction |
> | --- | --- | --- | --- |
> | Helped | $\Delta_s > 1\%$ | 725 | 70.8% |
> | Unchanged | $\|\Delta_s\| \leq 1\%$ | 246 | 24.0% |
> | Harmed | $\Delta_s < -1\%$ | 53 | 5.2% |
>
> Due to the space limit, key takeaways from the per-layer breakdown:
>
> The harmed heads concentrate in early layers (layers 3–6, 16.1% harmed) while mid-to-deep layers (7–18) show near-universal improvement (89.6% helped, 0.8% harmed). As analyzed in Appendix B.3, the negatively affected heads are predominantly those exhibiting the "Slash Line" pattern (Fig. 9a). **We will integrate this quantitative characterization into the main text of the revised paper.**
>
> Some more insights about the slash line pattern where permutation would harm block sparsity:
>
> Recent studies[1] observed that the slash-line pattern is closely related to RoPE, where the high-freq dimensions dominate and yield attention patterns that focus solely on tokens with fixed distances. An interesting connection to our results is that, the permutation gain for Qwen2.5-1M(RoPE base = 1M) is much higher than for Llama3.1(RoPE base = 500K). A higher base leads to fewer high-freq dimensions that could lead to slash-line patterns (potentially fewer slash-line heads), therefore benefiting more from permutation.
>
> [1] https://arxiv.org/abs/2601.08297
>
> **Response to W2/W3:**
>
> We completely agree that the final performance depends on the interaction between the permutation and the block scoring mechanism. Because permutation reduces information fragmentation at the structural level, we claim the benefit is agnostic to the selector. **We deliberately used the naive MeanPooling selector to *isolate* the structural benefits of permutation so that any observed gain is attributable to permutation alone, not selector heuristics.**
>
> When we state in Section 3.4 that the permutation gain is "agnostic to the specific block selection algorithm," we mean that **permutation fundamentally reduces the theoretical lower bound of blocks required to capture the attention mass (i.e., under oracle selection)**, ***any* selection algorithm trying to approximate the oracle will have a structurally easier target to hit.**
>
> We note that all block selection algorithms are essentially heuristics designed to simulate the oracle selection. Methods like MInference/FlexPrefill are explicitly tuned to approximate this oracle based on the spatial layout of *unpermuted* sequences (e.g., explicitly capturing slash lines). Because our permutation fundamentally restructures this spatial layout, these rigid, layout-dependent heuristics **no longer hold optimality for approximating the oracle selection** for the permuted sequence. Therefore, rather than forcing compatibility with heuristics tuned for the wrong layout, **we provide a systematic evaluation using general, layout-agnostic block selectors (MeanPooling and XAttention) that dynamically score blocks on the fly:**
>
> | Block Selector | Benchmark | w/o Permutation | w/ Permutation | Improvement ($\Delta$) |
> | --- | --- | --- | --- | --- |
> | MeanPooling | LongBench | 36.67 | 37.37 | +0.70 |
> | MeanPooling | RULER | 82.76 | 85.97  | +3.21 |
> | XAttention | LongBench | 36.42 | 36.80 | +0.38 |
> | XAttention | RULER | 86.46 | 86.87 | +0.41 |
>
> As shown above and in Appendix D.2 (Fig. 10), permutation increases block-level sparsity and improves task performance regardless of the selector.
>
> The RULER caveat (W3) is consistent with our claim: permutation densifies the matrix regardless of the selector, improving MeanPooling by +3.21 on RULER. When paired with a stronger selector like XAttention, the gains compound further, demonstrating the plug-and-play nature of permutation. We will integrate this analysis into the revised main text.
>
> **Response to Q1:**
>
> We acknowledge that the robustness of the last-query-block proxy is a common concern across reviewers. **We refer the reviewer to our response to Reviewer K92x (W1/Q1)**, where we provide a comprehensive ablation covering 5 proxy strategies across 8 diverse tasks. The key finding is that Random-Query-Attn, Avg-Query-Attn, and Last-Query-Block-Attn yield comparable sparsity gains across all tasks — including retrieval tasks where attention is highly query-specific. This confirms that the sparsity gain stems from clustering task-agnostic heavy hitters, not from the specific proxy choice or the attention pattern distribution of the task.

---

> > ### Author Rebuttal · Reviewer_8twW · 2026-04-04
> >
> > I thank the authors for their response, and I maintain my original score.

---

> > > ### Author Response · Authors · 2026-04-07
> > >
> > > Thank you for confirming all concerns are resolved.
> > > We put substantial effort into this rebuttal across all reviewers, and wanted to highlight the full picture:
> > >
> > > - New Qwen3-8B experiments (LongBench + efficiency) confirming generalizability.
> > > - 5-proxy × 8-task ablation showing the proxy choice is query-irrelevant and stable across domains.
> > > - Concrete head-level stats over all 1024 heads of Llama3.1-8B (70.8% helped, only 5.2% harmed), along with a connection to RoPE explaining why "slash-line" heads resist permutation.
> > > - Concrete block-count measurements from 8K–128K showing increasing reduction with length
> > > - Cross-selector comparison (MeanPooling + XAttention ± permutation).
> > > - Discussion on training-time compatibility with MoBA/NSA.
> > >
> > > We feel the paper is meaningfully stronger after rebuttal and hope you might take another look on these improvements. We totally understand if your view on the final score hasn't changed. Either way, we sincerely thank you for pushing us to do this rigorous work and for your help in improving the paper.

---

### Official Review · Reviewer_UnrX · 2026-03-11

**Soundness:** 3
**Presentation:** 4
**Significance:** 4
**Originality:** 4
**Overall Recommendation:** 5
**Confidence:** 4

**Summary:**

This paper proposes Permuted Block-Sparse Attention (PBS-Attn) for efficient LLM prefilling. The key idea is to exploit the permutation invariance of attention and reorder tokens within segments so that globally important keys become more concentrated in fewer blocks, improving block-level sparsity while preserving causal structure through a segmented permutation design. The method is presented as a plug-and-play enhancement on top of block-sparse attention and is implemented with custom permuted-FlashAttention kernels. Experiments on LongBench, LongBenchv2, and RULER show that PBS-Attn generally outperforms several sparse-attention baselines and achieves up to 2.75x end-to-end prefilling speedup while maintaining performance close to full attention

**Compliance With Llm Reviewing Policy:**

Affirmed.

**Final Justification:**

The rebuttal totally addresses my concerns. In particular, the added proxy ablations make the choice of the last-query-block proxy more convincing, and the new task-level and head-level analyses clarify better when permutation helps or hurts. The added selected-block statistics also address my request for a more concrete sparsity metric. Some uncertainty remains since part of the new evidence is only summarized in the rebuttal, but overall the paper is stronger after rebuttal. I will raise my score from 4 to 5.

**Key Questions For Authors:**

How sensitive is the method to the attention pattern distribution of the task? For example, does performance change on tasks where queries attend to different regions of the sequence?


Can the authors report metrics that directly measure block sparsity improvements (e.g., number of selected blocks before/after permutation)?


How sensitive is the method to the segment size used for permutation?

**Limitations:**

Yes

**Strengths And Weaknesses:**

Strengths
1. The paper introduces an interesting perspective: improving block sparsity via token permutation rather than designing new sparse selectors. This idea is conceptually simple and largely orthogonal to existing sparse attention approaches.

2. The work addresses an important practical problem (prefill efficiency for long-context LLMs) and demonstrates non-trivial engineering effort in implementing a permutation-aware attention kernel.

3. Empirical results suggest meaningful speedups while maintaining competitive accuracy.


Weaknesses
1. The key heuristic—ranking keys using similarity with the last query block—is insufficiently justified. It is unclear why a single query block should reliably represent the attention patterns of the entire sequence.

2. The experimental evaluation does not include ablations of the heuristic (e.g., random queries, averaged queries, or multiple query samples), making it difficult to determine whether the specific scoring strategy is essential.

3. The paper provides limited analysis of when the method works or fails. If attention patterns vary significantly across queries (e.g., retrieval or reasoning tasks), the proposed permutation may not capture global key importance.

4. The evaluation focuses on latency and accuracy but does not directly measure improvements in block sparsity or token concentration, which are central to the method’s motivation.

---

> ### Author Rebuttal · Authors · 2026-03-30
>
> We sincerely thank the reviewer for the thorough and constructive evaluation, and are glad that the reviewer recognizes PBS-Attn as "conceptually simple and largely orthogonal to existing sparse attention approaches" with "meaningful speedups while maintaining competitive accuracy." We address each concern below.
>
> **Response to W1/W2/W3/Q1:**
>
> We thank the reviewer for these closely related concerns. We address them jointly with a unified analysis spanning proxy justification, task-level sensitivity, and a granular characterization of when permutation helps vs. hurts.
>
> **Proxy justification and ablation (W1/W2).**
>
> We'd like to **first note that the robustness of the last-query-block proxy has been analyzed in our submission: in Sec. 2.2 and Fig. 1**, we compare it against using random query tokens and observe nearly identical sparsity gains, supporting our claim that the permutation benefit is query-agnostic. Additionally, **Fig. 3 visually demonstrates that the permutation effect is clustering the heavy-hitter vertical lines, rather than relying on query-specific information.**
>
> To further strengthen this, we conduct a comprehensive ablation covering 5 proxy strategies across 8 diverse tasks. **We refer the reviewer to our response to Reviewer K92x (W1/Q1) for the full table and detailed analysis due to space constraint.**
>
> In summary, Random-Query-Attn, Avg-Query-Attn (requested by the reviewer), and Last-Query-Block-Attn yield comparable sparsity gains across all tasks. This confirms that **the sparsity gain is query-irrelevant** and stems from clustering heavy hitters, not from the specific proxy choice.
>
> **When does permutation help vs. hurt? (W3/Q1).**
>
> This newly added ablation illustrates that **across all domains spanning Story QA (`narrativeqa`), Multi-hop QA (`hotpotqa`), Summarization (`gov_report`), Few-shot (`trec`), Code (`lcc`), and Retrieval (`niah_single/niah_multikey/vt`), permutation genuinely produces a noticeable sparsity gain within limited variance.** Notably, even retrieval tasks where queries attend to highly localized regions, show strong sparsity improvements (e.g., Δ_s = 4.69 for `niah_multikey`). This further strengthens our claim that **the efficacy of permutation is rather pattern-specific (learned behaviors of individual heads) than task-specific.** **We refer the reviewer to Appendix B.2 and B.3, where we provide a thorough head-level analysis and failure mode analysis.** Fig. 8 illustrates that for most of the heads, permutation leads to a noticeable sparsity gain. We further visualize the attention maps of heads with negative sparsity gains in Fig. 9, showing that permutation fails when the slash line pattern dominates the attention map, which only accounts for a minority of the total heads.
>
> Here we provide a head-level characterization of when permutation helps vs. hurts. We compute absolute sparsity gain $\Delta_s$ per head across all 32 layers of Llama-3.1-8B (1024 heads total) at 32K context and 97.5% coverage, measured on `niah_single` task:
>
> | Category  | Criterion          | Count | Fraction |
> |-----------|--------------------|-------|----------|
> | Helped    | $\Delta_s > 1\%$   |   725 |   70.8%  |
> | Unchanged | $\|\Delta_s\| \leq 1\%$ |   246 |   24.0%  |
> | Harmed    | $\Delta_s < -1\%$  |    53 |    5.2%  |
>
> **The results demonstrate that over 70% of heads benefit from permutation, while only 5.2% are negatively affected.**
>
> **Response to W4/Q2:**
>
> We thank the reviewer for this suggestion. **We have reported this metric in our submission.** Figure 5 directly measures the block-level density with and without permutation across context lengths from 8K to 128K at a fixed selection threshold of 0.9. To present this in concrete block counts as the reviewer requested, we provide the following breakdown (measured on Llama-3.1-8B with B=128, averaged across all layers and heads):
>
> | Context | Total Causal Blocks | Selected (w/o Perm) | Selected (w/ Perm) | Reduction | Rel. % |
> | --- | --- | --- | --- | --- | --- |
> | 8K | 2,080 | 1,350 | 1,205 | 145 | 10.7% |
> | 16K | 8,256 | 4,457 | 4,005 | 452 | 10.1% |
> | 32K | 32,896 | 15,135 | 13,337 | 1,798 | 11.9% |
> | 64K | 131,328 | 43,064 | 37,632 | 5,432 | 12.6% |
> | 128K | 524,800 | 134,270 | 114,996 | 19,274 | 14.4% |
>
> **Permutation consistently reduces the number of selected blocks, with the relative reduction scaling from 10.7% at 8K to 14.4% at 128K.** This confirms that structural optimization becomes increasingly effective for longer, more fragmented contexts, directly translating to the end-to-end speedups reported in Fig. 4.
>
> **Response to Q3:**
>
> **We have studied this in our submission. Figure 6b analyzes the effect of segment size $S$ on the performance-density trade-off.** A larger $S$ improves sparsity by clustering more tokens at once, but also enlarges the on-diagonal segments that cannot be skipped.

---

> > ### Author Rebuttal · Reviewer_UnrX · 2026-04-03
> >
> > The rebuttal largely addresses my main concerns. In particular, the added proxy ablations make the choice of the last-query-block proxy more convincing, and the new task-level and head-level analyses clarify better when permutation helps or hurts. The added selected-block statistics also directly address my request for a more concrete sparsity metric. Some uncertainty remains since part of the new evidence is only summarized in the rebuttal, but overall the paper is stronger after rebuttal. I will raise my score from 4 to 5.

---

> > > ### Author Response · Authors · 2026-04-03
> > >
> > > Thank you for the detailed feedbacks. We will incorporate all new evidence into the revised paper. Thank you for the effort in strengthening our paper.

---

### Official Review · Reviewer_K92x · 2026-03-13

**Soundness:** 3
**Presentation:** 3
**Significance:** 4
**Originality:** 4
**Overall Recommendation:** 5
**Confidence:** 5

**Summary:**

This paper addresses the computational and memory bottlenecks of the self-attention mechanism in Large Language Models (LLMs) when processing long contexts. The authors note that existing block-sparse attention methods often achieve sub-optimal block-level sparsity due to "information fragmentation". This fragmentation occurs when important key tokens for queries are scattered across multiple blocks, causing computational redundancy during block retrieval.

To resolve this, the authors propose Permuted Block-Sparse Attention (PBS-Attn), a plug-and-play method that leverages the permutation properties of attention to restructure the key sequence. Instead of passively selecting blocks, PBS-Attn actively groups globally important key tokens ("heavy hitters") into contiguous, high-density regions by sorting them based on a proxy global importance score. Furthermore, to maintain the strict causality required for LLM generation, the authors introduce a "Segmented Permutation" strategy, which applies permutations locally within segments while preserving inter-segment causal constraints.

**Key Contributions:**
* **Identification of Information Fragmentation:** The authors analyze the limitations of existing block-sparse methods, demonstrating that passively retrieving scattered "heavy hitter" tokens leads to diminished computational returns.
* **Permuted Block-Sparse Attention (PBS-Attn):** The introduction of a novel, plug-and-play sparse attention strategy that actively clusters globally important key tokens to significantly enhance block-level sparsity.
* **Segmented Permutation for Causality:** A theoretical and practical framework (Segmented Permutation) that preserves autoregressive causality by enforcing inter-segment causal masking while permitting intra-segment token rearrangement.
* **Empirical Validation and Speedup:** Comprehensive evaluations on long-context benchmarks (LongBench, LongBenchv2, and RULER) demonstrate that PBS-Attn matches the full-attention baseline accuracy while achieving up to a 2.75x end-to-end speedup in LLM prefilling.

**Compliance With Llm Reviewing Policy:**

Affirmed.

**Ethical Review Concerns:**

yes

**Final Justification:**

The authors have addressed most of my concerns. I maintain my positive score and expect the rebuttal's additional results and GQA discussion to be included in the final version.

**Key Questions For Authors:**

### Key Questions For Authors

**1. Robustness of Proxy Scoring to Localized Contexts (e.g., Needle In A Haystack)**
The proposed method relies heavily on using the last block of queries ($Q_{last\_block}$) to compute a proxy global importance score for key sorting. How does PBS-Attn perform on tasks requiring highly localized, non-global information retrieval—such as "Needle In A Haystack" tests—where the critical key token (the "needle") might have very low semantic similarity to the final query block? Have you observed any catastrophic forgetting or missing of local context?

**2. GQA Handling: Memory Overhead vs. Representation Power**
For models utilizing Grouped Query Attention (GQA), the default PBS-Attn strategy replicates keys and values to apply unique permutations per query head, which inherently negates the memory-saving benefits of GQA.While the shared permutation strategy is introduced in Appendix G, why wasn't this made the default? Furthermore, does this shared strategy degrade performance on complex reasoning or coding tasks where different heads within the same group might need to attend to vastly different attention patterns?

**Limitations:**

YES

**Strengths And Weaknesses:**

**1. Soundness**
The technical foundation and experimental design of the submission are generally robust, though there are a few implementation and methodological compromises worth noting.

* ***Strengths:***

1) Rigorous Theoretical Proofs: The authors provide solid theoretical foundations by mathematically proving the permutation invariance and equivariance of the attention mechanism.

2) Causality Preservation: They ingeniously resolve the critical issue of causality disruption in auto-regressive generation by introducing the "Segmented Permutation" strategy, which enforces inter-segment causal constraints while allowing intra-segment reordering.

3) Honest Failure Mode Analysis: The paper transparently analyzes its limitations, explicitly showing that the permutation method yields marginal or no sparsity gains for attention heads dominated by specific patterns (like "Slash Lines" or highly query-specific patterns).

4) Comprehensive Empirical Validation: The claims are well-supported by extensive experiments across various context lengths using modern LLMs on multiple long-context benchmarks.

* ***Weakness:***

Heuristic Proxy Scoring: The method relies on using only the last block of queries to estimate the global importance of all key tokens. While effective for identifying global "heavy hitters," this heuristic approach runs the risk of missing localized key tokens that are critical only to specific intermediate queries.

**2. Presentation**
The submission is clearly written, logically structured, and the core narrative is highly accessible.

* ***Strengths:***

1) Intuitive Problem Definition: The paper clearly articulates the limitations of existing block-sparse methods by identifying the "information fragmentation" problem and justifying the need for structural consolidation.

2) Reproducibility: The overall narrative is easy to follow, and the authors provide clear pseudocode for both the proposed mechanism and the baseline block selection logic.

3) Effective Visualizations: Visual aids, such as the coverage-density trade-off charts and attention map comparisons, effectively communicate the structural benefits of token permutation without relying solely on complex equations.

* ***Weakness:***

Relegation of GQA Discussion: Modern LLMs heavily rely on Grouped Query Attention (GQA). The paper's default strategy of replicating keys and values undermines GQA's memory efficiency, and the alternative "shared permutation" strategy is relegated entirely to the appendix. Because GQA is a standard architectural component today, this critical practical consideration should be integrated more prominently into the main text.


**3. Significance**

1) Addresses a Critical Bottleneck: The paper directly tackles the quadratic computational and memory complexities of the self-attention mechanism, which is a major hurdle for scaling long-context processing in LLMs.

2) High Practical Utility (Training-Free): The proposed method is a plug-and-play solution that can be applied to existing, pre-trained LLMs without the need for architectural changes or additional retraining, making it highly accessible for practitioners.

3) Meaningful Performance Acceleration: The method provides substantial performance improvements, achieving significant end-to-end speedups at extreme context lengths (e.g., up to 2.75x at 256K and over 3x attention speedup at 512K), all while maintaining accuracy levels close to the full-attention baseline.

**4. Originality**

1) Paradigm Shift in Sparse Attention: While prior block-sparse methods focus purely on passively selecting blocks from a fixed, scattered attention matrix, this work introduces the novel perspective of actively restructuring the matrix itself to inherently increase block-level sparsity.

2) Creative Combination Overcoming Constraints: The concept of token permutation has been explored in bidirectional models previously, but it was generally considered incompatible with the strict causal constraints of auto-regressive LLMs. The authors successfully bridge this gap and remove this restrictive assumption with their well-articulated segmented permutation framework.

---

> ### Author Rebuttal · Authors · 2026-03-30
>
> We sincerely thank the reviewer for the highly encouraging evaluation. We address each concern below:
>
> **Response to W1/Q1:**
>
> We acknowledge that the robustness of the last-query-block-attn proxy for token permutation is a common concern of the reviewers (`K92x` , `UnrX` and `8twW` ). Here we’d like to clarify that, as we emphasized in the paper(Sec. 2.2 and Fig. 1), **clustering global heavy hitters delivers higher density of the attention mass, and the last-query-block is only for identifying these heavy hitters.** Note that, heavy hitter tokens are key tokens that are critical to all query tokens (exhibiting a vertical line on the attention map), and **their existence is task-agnostic**. This is a known phenomenon of pretrained LLMs, also observed in well-established papers like H2O[1].  The reason we use the last block of queries is that this is the only block that all key blocks are visible to under the causal constraint, which makes it a natural choice for identifying heavy hitters. In Sec. 2.2, we have analyzed the robustness of this choice by comparing to using random query tokens for proxy, and observed nearly identical results, which confirms our “heavy hitter clustering” strategy. Another evidence is Fig. 3, which visually demonstrates that the permutation effect of the last-query-block proxy is clustering the vertical lines.
>
> **To prove this more rigorously, we further measure the sparsity gain from permutation on different tasks covering diverse domains,** **including:  `narrativeqa` (Story QA), `hotpotqa` (Multi-hop QA), `gov_report` (Summarization), `trec` (Few-shot), `lcc` (Code),  `niah_single/niah_multikey` (Retrieval), `vt` (Variable Tracing).**
>
> | Method | narrativeqa | hotpotqa | gov_report | trec | lcc | niah_single | niah_multikey | vt |
> | --- | --- | --- | --- | --- | --- | --- | --- | --- |
> | Random | -6.11 | -9.07 | -6.67 | -7.38 | -7.45 | -6.87 | -5.31 | -7.53 |
> | Greedy | 3.00 | 0.74 | 1.19 | 1.62 | 1.70 | 0.78 | 1.32 | 0.61 |
> | Random Query Attn | 7.90 | 1.30 | 3.19 | 5.39 | 6.97 | 3.25 | 4.69 | 4.50 |
> | Avg Query Attn | 7.73 | 2.35 | 4.22 | 4.39 | 8.02 | 3.35 | 4.79 | 4.48 |
> | Last Query Block Attn | 7.77 | 0.41 | 2.82 | 5.25 | 7.42 | 3.34 | 4.72 | 4.69 |
>
> **The table illustrates the absolute sparsity gain($\Delta_s$) across different permutation proxies covering diverse domains at a attention coverage of 97.5% with oracle block selection.** The results are consistent with those in Fig. 1:
>
> 1. Random Permutation disrupts attention sparsity;
> 2. Allocating key blocks using local query centroids (Greedy) yields suboptimal sparsity gain compared to clustering global critical key tokens;
> 3. Random/Avg/Last-Block-Query-Attn yields similar sparsity gain, demonstrating that the source of the permutation sparsity gain is query-irrelevant, which is clustering the heavy hitters.
>
> We will add this task-wise sensitivity analysis to the revised paper to strengthen the justification of our proxy mechanism.
>
> To directly answer the reviewer’s question, we believe that it’s crucial to clarify the division of labor in our method: **the proxy is only used for the Permutation Phase to group global heavy hitters into dense blocks; it is *not* a pruning mechanism.** If a highly localized key (a "needle") receives a low global proxy score, it simply remains unpromoted in the KV cache layout. During the **Block Selection Phase**, the actual query evaluates local similarity on-the-fly and retrieves the needle's block normally. Permutation only densifies the attention mass to reduce redundant computation. As seen in the table, permutation consistently improves the sparsity gain for NIAH tasks. Any catastrophic forgetting or missing of local context should only be due to the under selection of blocks, which should also happen to the unpermuted sequence.
>
> [1] https://arxiv.org/abs/2306.14048
>
> **Response to W2/Q2:**
>
> We use unique permutation for each head in the main paper for 3 reasons:
>
> 1. Our primary objective was to establish the theoretical upper bound of absolute block-level sparsity and latency speedup. Since assigning a unique permutation to each query head naturally yields the highest possible sparsity, we evaluated it as our default.
> 2. The prefilling phase is compute-bound rather than memory-bound, the memory-saving benefit of GQA is less critical during the compute-bound prefilling phase. Since in PBS-Attn, the permuted KVs are not saved for later decoding, it won’t diminish the benefit of GQA in decoding.
> 3. As we analyzed in Appendix E, the memory overhead of permutation for each head is insignificant.
>
> However, the degradation of sharing permutation within groups is remarkably minimal. **As demonstrated in Appendix G (Table 6) of our submission, sharing the permutations only shows a marginal drop from 37.37 to 37.21 on LongBench, which includes complex reasoning tasks (e.g., `hotpotqa` ) and coding tasks (e.g., `lcc`).** We will integrate this GQA discussion into the revised main text.

---

> > ### Author Rebuttal · Reviewer_K92x · 2026-04-04
> >
> > The authors have addressed most of my concerns. I maintain my positive score and expect the rebuttal's additional results and GQA discussion to be included in the final version.

---

> > > ### Author Response · Authors · 2026-04-07
> > >
> > > Thank you for confirming that the concerns are addressed. We will incorporate all additional results and the GQA discussion into the final version as promised. Thank you again for your time and effort.

---

### Official Review · Reviewer_rmif · 2026-03-13

**Soundness:** 3
**Presentation:** 3
**Significance:** 3
**Originality:** 3
**Overall Recommendation:** 4
**Confidence:** 3

**Summary:**

This paper proposes PBS-Attn, a plug-and-play block sparse attention method that accelerates long-context prefilling by permuting tokens to increase block-level sparsity. The key idea is to cluster globally important tokens into denser blocks through segmented permutation, while preserving causal structure. Unlike existing block sparse attention methods mainly focusing on block selection, this paper optimizes the attention structure structure itself for better block selection. The proposed method is training-free, and the empirical results show consistently better quality than prior block-sparse baselines and up to 2.75× end-to-end prefilling speedup.

**Compliance With Llm Reviewing Policy:**

Affirmed.

**Final Justification:**

PBS-Attn presents a novel idea of improving block sparsity through token permutation, which is well-motivated and practically useful. The method is training-free, technically sound, and supported by strong empirical results and solid systems engineering.

My main concern regarding limited model evaluation has been fully addressed by the rebuttal, with new Qwen3-8B results confirming generalizability and a helpful pointer to the existing larger-model evaluations in Appendix F. The discussion on compatibility with training-time methods was also thoughtful.

I keep my score at 4 (weak accept). While the rebuttal resolved my concerns, the paper remains a solid but incremental contribution to efficient long-context inference that others are likely to build on.

**Key Questions For Authors:**

1. Can this method be combined with training-time sparse attention methods like MoBA or NSA?

**Limitations:**

Yes.

**Strengths And Weaknesses:**

**Strengths**
1. Novelty.
The paper introduces a novel and interesting idea: improving block sparsity through token permutation rather than only designing better selection method. The segmented permutation design is also a reasonable way to preserve causality.
2. Good practical value.
The method is training-free and tackles an important problem in LLM inference, the quadratic complexity of attention during prefilling.
3. Strong empirical results.
The method consistently outperforms prior sparse attention baselines on multiple long-context benchmarks, while remaining close to full attention accuracy. The reported speedup is strong.
4. Good systems effort.
The custom permuted-FlashAttention kernel is solid and makes the speedup results more convincing.

Weaknesses
1. Only evaluated on limited models.
The results are promising, but it would be helpful to see whether the same benefits remain strong on latest models like Qwen3 or other larger models.

---

> ### Author Rebuttal · Authors · 2026-03-30
>
> **Response to W1:**
>
> We thank the reviewer for raising this. To demonstrate the generalizability of PBS-Attn, we evaluate the more recent Qwen3-8B on LongBench. All hyperparameters are consistent with those in the paper:
>
> | Method | Single-Doc QA | Multi-Doc QA | Summarization | Few-Shot | Code | Synthetic | Overall |
> | --- | --- | --- | --- | --- | --- | --- | --- |
> | FlashAttn | 46.93 | 37.97 | 16.57 | 34.07 | 2.93 | 66 | 34.08 |
> | Minference | 46.92 | 37.11 | 16.53 | 34.15 | 2.98 | 66.17 | 33.98 |
> | FlexPrefill | 46.08 | 37.29 | 16.53 | 33.63 | 2.52 | 49 | 30.84 |
> | XAttention | 45.66 | 37.77 | 16.66 | 36.2 | 2.02 | 64.67 | 33.83 |
> | MeanPool | 45.69 | 37.25 | 16.61 | 37.45 | 2.52 | 62.33 | 33.64 |
> | PBS-Attn | 47.04 | 37.17 | 16.66 | 33.88 | 2.8 | 66.33 | 33.98 |
>
> **The results demonstrate that PBS-Attn seamlessly generalizes to Qwen3-8B, maintaining comparable performance to full attention.** We further validate the efficiency gain on Qwen3-8B(e2e speedup, measured in ms):
>
> | Method | 8k | 16k | 32k | 64k | 128k | 256k(tp=4) | 512k(tp=8) |
> | --- | --- | --- | --- | --- | --- | --- | --- |
> | FlashAttn | 326.8 | 732.7 | 1905.3 | 5573 | 18628 | 20104.7 | 53999.5 |
> | PBS-Attn | 368.9 | 715 | 1536.4 | 3540.4 | 8396 | 7400.9 | 28136.1 |
> | Speedup | 0.89x | 1.02x | 1.24x | 1.57x | 2.22x | 2.72x | 1.92x |
>
> **These speedup results are nearly identical to the strong scaling trajectory observed on Llama-3.1-8B in our main paper.** As noted in Section 4.2, the 256K and 512K evaluations employ tensor parallelism (tp=4 and tp=8, respectively) to accommodate memory constraints. While the communication overhead of tensor parallelism naturally limits the theoretical maximum *end-to-end* speedup of any sparse attention method at extreme lengths, PBS-Attn still delivers a highly practical 2.72× speedup at 256K.
>
> **Regarding the evaluation on larger models, we kindly refer the reviewer to Appendix F (Table 5 and Figure 12) of our submission.** There, we detail comprehensive performance and efficiency evaluations on the **Qwen-2.5-14B-1M** model. The results confirm that PBS-Attn maintains its superior performance and achieves nearly identical E2E speedups on larger architectures, proving the scalability of our approach.
>
> We will include the Qwen3-8B results in the revised paper and add a more prominent pointer to the larger model evaluations (Appendix F) in the main text.
>
> **Response to Q1:**
>
> We thank the reviewer for this insightful question. **Yes, PBS-Attn is fundamentally orthogonal to and highly synergistic with training-time sparse attention methods like MoBA and NSA.**
>
> While methods like MoBA and NSA focus on optimizing the *routing mechanism* (i.e., learning to select the most relevant key blocks), **their efficiency remains bottlenecked by the inherently scattered distribution of the KV cache, which is the fundamental bottleneck of block selection algorithms as we discussed in the paper.** By applying PBS-Attn to physically densify the attention mass into fewer blocks, these training-time routers would be able to perform thriftier and more accurate block selection. **This could lead to faster convergence of the routing optimization and allow higher accuracy under a significantly reduced top-k compute budget.**
>
> Theoretically, the permutation properties of attention (Theorem 1) hold equally for training-time methods, and segmented permutation remains essential for causal language model training. One consideration is that our current permutation criterion (global permutation for heavy hitters) is specifically designed based on the pretrained attention patterns and evaluated on training-free settings, it’s not clear whether this approach is optimal for permutation in training-time methods. Nevertheless, it is a promising direction to apply PBS-Attn to training-time sparse methods, maybe with minor adaptations on the specific permutation criterion. Thanks again for pointing this! We will add a dedicated discussion to the revised paper.

---

> > ### Author Rebuttal · Reviewer_rmif · 2026-04-02
> >
> > The rebuttal addressed my concerns, and I am satisfied with the clarifications and additional experiments. I will keep my positive rating.

---

> > > ### Author Response · Authors · 2026-04-02
> > >
> > > Thank you for confirming that all concerns have been fully resolved. Given this, we respectfully ask whether you might consider adjusting your score, as a score of 4 indicates "with some weaknesses that limit its impact," which may no longer apply now that your concerns have been addressed with additional experiments. Thank you for your time and effort.

---

### Decision · Program_Chairs · 2026-04-30

**Decision:**

Accept (regular)

**Comment:**

Developing efficient sparse attention is fundamental to scaling LLMs to long contexts. The paper proposes PBS-Attn, a sparse attention method that permutes tokens to cluster important keys into fewer blocks, with segmented permutation preserving causality.

The method is backed by theory and strong results (up to 2.75 $\times$ speedup at 256K with accuracy near full attention). All four reviewers are positive.

The main concern was the justification of the last-query-block proxy and limited evidence of when permutation helps vs. hurts.
The authors addressed this with a 5-proxy $\times$ 8-task ablation, Qwen3-8B experiments, and cross-selector comparisons. Hope the authors incorporate these into the final revision.